# Your Neighbors Are Communicating: Towards Powerful and Scalable Graph Neural Networks

## Abstract

Message passing graph neural networks (GNNs) are known to have their expressiveness upper-bounded by 1-dimensional Weisfeiler-Lehman (1-WL) algorithm. To achieve more powerful GNNs, existing attempts either require *ad hoc* features, or involve operations that incur high time and space complexities. In this work, we propose a *general* and *provably powerful* GNN framework that preserves the *scalability* of message passing scheme. In particular, we first propose to empower 1-WL for graph isomorphism test by considering edges among neighbors, giving rise to NC-1-WL. The expressiveness of NC-1-WL is shown to be strictly above 1-WL and below 3-WL theoretically. Further, we propose the NC-GNN framework as a differentiable neural version of NC-1-WL. Our simple implementation of NC-GNN is provably as powerful as NC-1-WL. Experiments demonstrate that our NC-GNN achieves remarkable performance on various benchmarks.

## 1 Introduction

Graph Neural Networks (GNNs) (Gori et al., 2005; Scarselli et al., 2008) have been demonstrated to be effective for various graph tasks. In general, modern GNNs employ a message passing mechanism where the representation of each node is recursively updated by aggregating representations from its neighbors (Atwood & Towsley, 2016; Li et al., 2016; Kipf & Welling, 2017; Hamilton et al., 2017; Veličković et al., 2018; Xu et al., 2019; Gilmer et al., 2017). Such message passing GNNs, however, have been shown to be *at most* as powerful as the 1-dimensional Weisfeiler-Lehman (1-WL) algorithm (Weisfeiler & Lehman, 1968) in distinguishing non-isomorphic graphs (Xu et al., 2019; Morris et al., 2019). Thus, message passing GNNs cannot distinguish some simple graphs and cannot detect certain important structural concepts (Chen et al., 2020; Arvind et al., 2020).

Recently, a lot of efforts have been made to improve the expressiveness of message passing GNNs by considering high-dimensional WL algorithms (*e.g.*, Morris et al. (2019); Maron et al. (2019)), exploiting subgraph information (*e.g.*, Bodnar et al. (2021a); Zhang & Li (2021)), or adding more distinguishable features (*e.g.*, Murphy et al. (2019); Bouritsas et al. (2022)). As thoroughly discussed in Section 5, these existing methods either rely on handcrafted/predefined/domain-specific features, or require high computational cost and memory budget. In contrast, our goal in this work is to **develop a *general* GNN framework with *provably expressive* power, while maintaining the *scalability* of the message passing scheme**.

Specifically, we first propose an extension of the 1-WL algorithm, namely NC-1-WL, by considering the edges among neighbors. In other words, we incorporate the information of which two neighbors are communicating (*i.e.*, connected) into the graph isomorphism test algorithm. To achieve this, we mathematically model the edges among neighbors as a *multiset of multisets*, in which each edge is represented as a *multiset* of two elements. We theoretically show that the expressiveness of our NC-1-WL in distinguishing non-isomorphic graphs is strictly above 1-WL and below 3-WL. Further, based on NC-1-WL, we propose a general GNN framework, known as NC-GNN, which can be considered as a differentiable neural version of NC-1-WL. We provide a simple implementation of NC-GNN that is proved to be as powerful as NC-1-WL. Compared to existing expressive GNNs, our NC-GNN is a general, provably powerful and, more importantly, scalable framework.

The main question addressed in our work is how to make best use of information in the one-hop neighborhood to improve expressive power while preserving scalability. In the one-hop neighborhood of each node, the local patterns we can consider are (A) what are the neighbors and (B) how the neighbors are connected to each other. The previous message passing GNNs only consider (A). We move a significant step forward to consider (B) by modeling edges among neighbors as a multiset of multisets, thereby leading to provably expressive power and preserved scalability. From this perspective, our method is fundamentally different from existing methods that encode triangle features, such as MotifNet (Monti et al., 2018) and SIGN (Rossi et al., 2020). Specifically, these methods employ triangle-related motif-induced adjacency matrices in their convolution and diffusion operators, respectively. The edge weight in a motif-induced adjacency matrix is obtained by multiplying the original edge weight with the frequency that each edge participates in triangle motifs. Compared to this hand-crafted way, our method is a general framework to encode how the neighbors are connected to each other, and the expressiveness of our framework can be rigorously characterized.

We perform experiments on graph classification and node classification to evaluate NC-GNN comprehensively. Our NC-GNN consistently outperforms GIN, which is as powerful as 1-WL, by significant margins on various tasks. Remarkably, NC-GNN outperforms GIN by an absolute margin over 12.0 on CLUSTER in term of test accuracy. In addition, NC-GNN performs competitively, often achieves better results, compared to existing expressive GNNs, while being much more efficient.

## 2  PRELIMINARIES

We start by introducing notations. We represent an undirected graph as $\mathcal{G} = (V, E, \boldsymbol{X})$, where $V$ is the set of nodes and $E \subseteq V \times V$ denotes the set of edges. We represent an edge $\{v, u\} \in E$ by $(v, u)$ or $(u, v)$ for simplicity. $\boldsymbol{X} = [\boldsymbol{x}_1, \cdots, \boldsymbol{x}_n]^T \in \mathbb{R}^{n \times d}$ is the node feature matrix, where $n = |V|$ is the number of nodes and $\boldsymbol{x}_v \in \mathbb{R}^d$ represents the $d$-dimensional feature of node $v$. $\mathcal{N}_v = \{u \in V | (v, u) \in E\}$ is the set of neighboring nodes of node $v$. A multiset is denoted as $\{\{\cdots\}\}$ and formally defined as follows.

**Definition 1** (Multiset). A *multiset* is a generalized concept of set allowing repeating elements. A multiset $X$ can be formally represented by a 2-tuple as $X = (S_X, m_X)$, where $S_X$ is the underlying set formed by the distinct elements in the multiset and $m_X : S_X \to \mathbb{Z}^+$ gives the *multiplicity* (*i.e.*, the number of occurrences) of the elements. If the elements in the multiset are generally drawn from a set $\mathcal{X}$ (*i.e.*, $S_X \subseteq \mathcal{X}$), then $\mathcal{X}$ is the *universe* of $X$ and we denote it as $X \subseteq \mathcal{X}$ for ease of notation.

**Message passing GNNs**. Modern GNNs usually follow a message passing scheme to learn node representations in graphs (Gilmer et al., 2017). To be specific, the representation of each node is updated iteratively by aggregating the multiset of representations formed by its neighbors. In general, the $\ell$-th layer of a message passing GNN can be expressed as

$$\boldsymbol{a}_v^{(\ell)} = f_{\text{aggregate}}^{(\ell)}\Big(\{\{\boldsymbol{h}_u^{(\ell-1)}|u \in \mathcal{N}_v\}\}\Big), \quad \boldsymbol{h}_v^{(\ell)} = f_{\text{update}}^{(\ell)}\Big(\boldsymbol{h}_v^{(\ell-1)}, \boldsymbol{a}_v^{(\ell)}\Big). \tag{1}$$

$f_{\text{aggregate}}^{(\ell)}$ and $f_{\text{update}}^{(\ell)}$ are the parameterized functions of the $\ell$-th layer. $\boldsymbol{h}_v^{(\ell)}$ is the representation of node $v$ at the $\ell$-th layer and $\boldsymbol{h}_v^{(0)}$ can be initialized as $\boldsymbol{x}_v$. After employing $L$ such layers, the final representation $\boldsymbol{h}_v^{(L)}$ can be used for prediction tasks on each node $v$. For graph-level problems, a graph representation $\boldsymbol{h}_G$ can be obtained by applying a readout function as,

$$\boldsymbol{h}_G = f_{\text{readout}}\Big(\{\{\boldsymbol{h}_v^{(L)}|v \in V\}\}\Big). \tag{2}$$

**Definition 2** (Isomorphism). Two graphs $\mathcal{G} = (V, E, \boldsymbol{X})$ and $\mathcal{H} = (P, F, \boldsymbol{Y})$ are *isomorphic*, denoted as $\mathcal{G} \simeq \mathcal{H}$, if there exists a *bijective* mapping $g : V \to P$ such that $\boldsymbol{x}_v = \boldsymbol{y}_{g(v)}, \forall v \in V$ and $(v, u) \in E$ iff $(g(v), g(u)) \in F$. Graph isomorphism is still an open problem without a known polynomial-time solution.

**Weisfeiler-Lehman algorithm**. The Weisfeiler-Lehman algorithm (Weisfeiler & Lehman, 1968) provides a hierarchy for graph isomorphism testing problem. Its 1-dimensional form (*a.k.a.*, 1-WL or color refinement) is a heuristic method that can efficiently distinguish a broad class of non-isomorphic graphs (Babai & Kucera, 1979). 1-WL assigns a color $c_v^{(0)}$ to each node $v$ according to its initial label

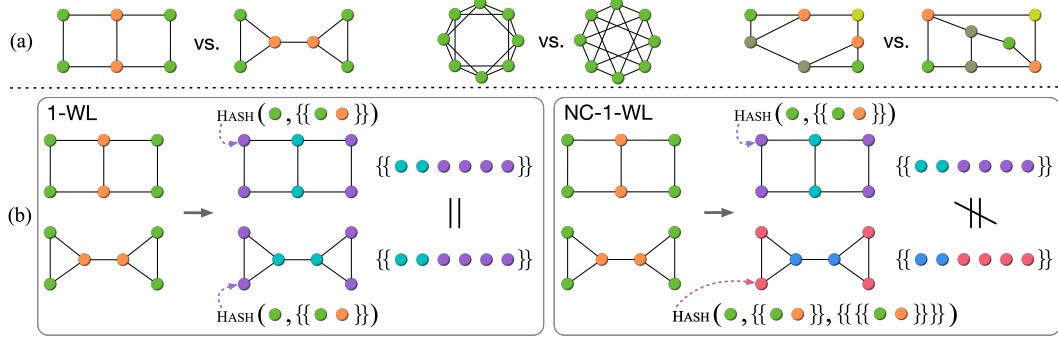

Figure 1: (a) Several example pairs of non-isomorphic graphs, partially adapted from Sato (2020), that cannot be distinguished by 1-WL. Colors represent initial node labels or features. Our NC-1-WL can distinguish them. (b) A comparison between the executions of 1-WL and NC-1-WL on two non-isomorphic graphs.

(or feature)[1] and then iteratively refines the colors until convergence. Convergence means that the subsets of nodes with the same colors can not be further split to different color groups. In particular, at each iteration $\ell$, it aggregates the colors of nodes and their neighborhoods, which are represented as multisets, and hashes the aggregated results into unique new colors (*i.e.*, injectively). Formally,

$$c_v^{(\ell)} \leftarrow \text{HASH}\Big(c_v^{(\ell-1)}, \{\{c_u^{(\ell-1)} | u \in \mathcal{N}_v\}\}\Big). \tag{3}$$

1-WL decides two graphs to be non-isomorphic once the colorings between these two graphs differ at some iteration. Instead of coloring each node, $k$-WL generalizes 1-WL by coloring each $k$-tuple of nodes and thus needs to refine the colors for $n^k$ tuples. The details of $k$-WL are provided in Algorithm 2, Appendix A.2. It is known that 1-WL is as powerful as 2-WL in terms of distinguishing non-isomorphic graphs (Cai et al., 1992; Grohe & Otto, 2015; Grohe, 2017). Moreover, for $k \geq 2$, $(k + 1)$-WL is strictly more powerful than $k$-WL[2] (Grohe & Otto, 2015). More details of the WL algorithms are given in Cai et al. (1992); Grohe (2017); Sato (2020); Morris et al. (2021).

Given the similarity between message passing GNNs and 1-WL algorithm (*i.e.* Eq. (1) *vs.* Eq. (3)), message passing GNNs can be viewed as a differentiable neural version of 1-WL. In fact, it has been shown that message passing GNNs are *at most* as powerful as 1-WL in distinguishing non-isomorphic graphs (Xu et al., 2019; Morris et al., 2019). Further, Xu et al. (2019) proves that message passing GNNs can achieve the same expressiveness as 1-WL if the *aggregate*, *update*, and *readout* functions are injective, thereby developing the GIN model (Xu et al., 2019). Thus, the expressive power of message passing GNNs is upper bounded by 1-WL. In other words, if two non-isomorphic graphs cannot be distinguished by 1-WL, then message passing GNNs must yield the same embedding for them. Importantly, such expressive power is not sufficient to distinguish some common graphs and cannot capture certain basic structural information such as triangles (Chen et al., 2020; Arvind et al., 2020), which play significant roles in certain tasks, such as tasks over social networks. Several examples that cannot be distinguished by 1-WL or message passing GNNs are shown in Figure 1 (a).

## 3   THE NC-1-WL ALGORITHM

In this section, we introduce the proposed NC-1-WL algorithm, which extends the 1-WL algorithm by taking the edges among neighbors into consideration. With such simple but non-trivial extension, NC-1-WL is proved to be strictly more powerful than 1-WL and less powerful than 3-WL, while preserving the efficiency of 1-WL.

---

[1]If there are no initial features or labels, 1-WL assigns the same color to all the nodes in the graph.

[2]There are two families of WL algorithms; they are $k$-WL and $k$-FWL (Folklore WL). They both consider coloring $k$-tuples and their difference lies in how to aggregate colors from neighboring $k$-tuples. It is known that $(k - 1)$-FWL is as powerful as $k$-WL for $k \geq 3$ (Grohe & Otto, 2015; Grohe, 2017; Maron et al., 2019). To avoid ambiguity, in this work, we only involve $k$-WL.

---

**Algorithm 1** NC-1-WL *vs.* 1-WL for graph isomorphism test

---

**Input:** Two graphs $\mathcal{G} = (V, E, \boldsymbol{X})$ and $\mathcal{H} = (P, F, \boldsymbol{Y})$

$c_v^{(0)} \leftarrow \text{HASH}(\boldsymbol{x}_v), \forall v \in V$

$d_p^{(0)} \leftarrow \text{HASH}(\boldsymbol{y}_p), \forall p \in P$

**repeat** $(\ell = 1, 2, \cdots)$

    **if** $\{\{c_v^{(\ell-1)} | v \in V\}\} \neq \{\{d_p^{(\ell-1)} | p \in P\}\}$ **then**

        **return** $\mathcal{G} \not\simeq \mathcal{H}$

    **end if**

    **for** $v \in V$ **do**

        $c_v^{(\ell)} \leftarrow \text{HASH}\Big(c_v^{(\ell-1)}, \{\{c_u^{(\ell-1)} | u \in \mathcal{N}_v\}\}, \underline{\{\{\{\{c_{u_1}^{(\ell-1)}, c_{u_2}^{(\ell-1)}\}\} | u_1, u_2 \in \mathcal{N}_v, (u_1, u_2) \in E\}\}}\Big)$

    **end for**

    **for** $p \in P$ **do**

        $d_p^{(\ell)} \leftarrow \text{HASH}\Big(d_p^{(\ell-1)}, \{\{d_q^{(\ell-1)} | q \in \mathcal{N}_p\}\}, \underline{\{\{\{\{d_{q_1}^{(\ell-1)}, d_{q_2}^{(\ell-1)}\}\} | q_1, q_2 \in \mathcal{N}_p, (q_1, q_2) \in F\}\}}\Big)$

    **end for**

**until** convergence

**return** $\mathcal{G} \simeq \mathcal{H}$

---

As shown in Eq. (3) and (1), 1-WL and message passing GNNs consider neighbors of each node as a multiset of representations. Here, we move one step forward by further treating edges among neighbors as a multiset, where each element is also a multiset corresponding to an edge. We formally define a multiset of multisets as follows.

**Definition 3** (Multiset of multisets). A *multiset of multisets*, denoted by $W$, is a multiset where each element is also a multiset. In this work, we only need to consider that each element in $W$ is a multiset formed by 2 elements. Following our definition of multiset, if these 2 elements are generally drawn from a set $\mathcal{X}$, the *universe* of $W$ is the set $\mathcal{W} = \{\{\{w_1, w_2\}\} | w_1, w_2 \in \mathcal{X}\}$. We can formally represent $W = (S_W, m_W)$, where the underlying set $S_W \subseteq \mathcal{W}$ and $m_W : S_W \to \mathbb{Z}^+$ gives the *multiplicity*. Similarly, we have $W \subseteq \mathcal{W}$.

Particularly, our NC-1-WL considers modeling edges among neighbors as a multiset of multisets and extends 1-WL (*i.e.*, Eq. (3)) to

$$c_v^{(\ell)} \leftarrow \text{HASH}\Big(c_v^{(\ell-1)}, \{\{c_u^{(\ell-1)} | u \in \mathcal{N}_v\}\}, \underbrace{\{\{\{\{c_{u_1}^{(\ell-1)}, c_{u_2}^{(\ell-1)}\}\} | u_1, u_2 \in \mathcal{N}_v, (u_1, u_2) \in E\}\}}_{\textit{A multiset of multisets}}\Big).$$

$$(4)$$

As 1-WL, our NC-1-WL determines two graphs to be non-isomorphic as long as the colorings of these two graphs are different at some iteration. We summarize the overall process of NC-1-WL in Algorithm 1, where the difference with 1-WL is underlined.

Importantly, our NC-1-WL is more powerful than 1-WL in distinguishing non-isomorphic graphs. Several examples that cannot be distinguished by 1-WL are shown in Figure 1 (a). Our NC-1-WL can distinguish them easily. An example of executions is demonstrated in Figure 1 (b). We rigorously characterize the expressiveness of NC-1-WL by the following theorems. The proofs are given in Appendix A.1 and A.2.

**Theorem 1.** *NC-1-WL is strictly more powerful than 1-WL in distinguishing non-isomorphic graphs.*

**Theorem 2.** *NC-1-WL is strictly less powerful than 3-WL in distinguishing non-isomorphic graphs.*

Although NC-1-WL is less powerful than 3-WL, it is much more efficient. 3-WL has to refine the color of each 3-tuple, resulting in $n^3$ refinement steps in each iteration. In contrast, as 1-WL, NC-1-WL only needs to color each node, which corresponds to $n$ refinement steps in each iteration. Thus, the superiority of our NC-1-WL lies in improving the expressiveness over 1-WL, while being efficient as 1-WL.

Note that our NC-1-WL differs from the concept of Subgraph-1-WL (Zhao et al., 2022), which *ideally* generalizes 1-WL from mapping the neighborhood to mapping the subgraph rooted at each node. Specifically, the refinement step in Subgraph-1-WL is $c_v^{(\ell)} \leftarrow \text{HASH}\left(\mathcal{G}[\mathcal{N}_v^k]\right)$, where $\mathcal{G}[\mathcal{N}_v^k]$

is the subgraph induced by the $k$-hop neighbors of node $v$. However, it requires an injective hash function for subgraphs, which is essentially as hard as the graph isomorphism problem and cannot be achieved. In contrast, our NC-1-WL does not aim to injectively map the neighborhood subgraph. Instead, we enhance 1-WL by mathematically modeling the edges among neighbors as a multiset of multisets. Then, injectively mapping such multiset of multisets in NC-1-WL is naturally satisfied.

## 4 THE NC-GNN FRAMEWORK

In this section, we propose the NC-GNN framework as a differentiable neural version of NC-1-WL. Further, we establish an instance of NC-GNN that is provably as powerful as NC-1-WL in distinguishing non-isomorphic graphs.

Differing from previous message passing GNNs as Eq. (1), NC-GNN further considers the edges among neighbors as NC-1-WL. One layer of the NC-GNN framework can be formulated as

$$
\begin{aligned}
\boldsymbol{c}_v^{(\ell)} &= f_{\text{communicate}}{}^{(\ell)}\Big(\{\{\{\{\boldsymbol{h}_{u_1}^{(\ell-1)}, \boldsymbol{h}_{u_2}^{(\ell-1)}\}\}|u_1, u_2 \in \mathcal{N}_v, (u_1, u_2) \in E\}\}\Big), \\
\boldsymbol{a}_v^{(\ell)} &= f_{\text{aggregate}}{}^{(\ell)}\Big(\{\{\boldsymbol{h}_u^{(\ell-1)}|u \in \mathcal{N}_v\}\}\Big), \\
\boldsymbol{h}_v^{(\ell)} &= f_{\text{update}}{}^{(\ell)}\Big(\boldsymbol{h}_v^{(\ell-1)}, \boldsymbol{a}_v^{(\ell)}, \boldsymbol{c}_v^{(\ell)}\Big).
\end{aligned}
\tag{5}
$$

$f_{\text{communicate}}{}^{(\ell)}$ is the parameterized function operating on multisets of multisets. The following theorem establishes the conditions under which our NC-GNN can be as powerful as NC-1-WL.

**Theorem 3.** *Let $\mathcal{M} : \mathcal{G} \to \mathbb{R}^d$ be an NC-GNN model with a sufficient number of layers following Eq. (5). $\mathcal{M}$ is as powerful as NC-1-WL in distinguishing non-isomorphic graphs if the following conditions hold: (1) At each layer $\ell$, $f_{communicate}{}^{(\ell)}$, $f_{aggregate}{}^{(\ell)}$, and $f_{update}{}^{(\ell)}$ are injective. (2) The final readout function $f_{readout}$ is injective.*

The proof is provided in Appendix A.3. One may wonder *what advantages NC-GNN has over NC-1-WL*. Note that NC-1-WL only yields different colors to distinguish nodes according to their neighbors and edges among neighbors. These colors, however, do not represent any similarity information and are essentially one-hot encodings. In contrast, NC-GNN, a neural generalization of NC-1-WL, aims at representing nodes in the embedding space. Thus, an NC-GNN model satisfying Theorem 3 can not only distinguish nodes according to their neighbors and edges among neighbors, but also learn to map nodes with certain structural similarities to similar embeddings, based on the supervision from the on-hand task. This has the same philosophy as the relationship between message passing GNN and 1-WL (Xu et al., 2019).

There could exist many ways to implement the *communicate*, *aggregate*, and *update* functions in the NC-GNN framework. Here, following the NC-GNN framework, we provide a simple architecture, that provably satisfies Theorem 3 and thus has the same expressive power as NC-1-WL. To achieve this, we generalize the prior results of parameterizing universal functions over *sets* (Zaheer et al., 2017) and *multisets* (Xu et al., 2019) to consider both *multisets* and *multisets of multisets*. Such non-trivial generalization is formalized in the following lemmas. The proofs are available in Appendix A.4 and A.5. As Xu et al. (2019), we assume that the node feature space is countable.

**Lemma 4.** *Assume $\mathcal{X}$ is countable. There exist two functions $f_1$ and $f_2$ so that $h(X, W) = \sum_{x \in X} f_1(x) + \sum_{\{\{w_1, w_2\}\} \in W} f_2(f_1(w_1) + f_1(w_2))$ is unique for any distinct pair of $(X, W)$, where $X \subseteq \mathcal{X}$ is a multiset with a bounded cardinality and $W \subseteq \mathcal{W} = \{\{\{w_1, w_2\}\}|w_1, w_2 \in \mathcal{X}\}$ is a multiset of multisets with a bounded cardinality. Moreover, any function $g$ on $(X, W)$ can be decomposed as $g(X, W) = \phi\big(\sum_{x \in X} f_1(x) + \sum_{\{\{w_1, w_2\}\} \in W} f_2(f_1(w_1) + f_1(w_2))\big)$ for some function $\phi$.*

**Lemma 5.** *Assume $\mathcal{X}$ is countable. There exist two functions $f_1$ and $f_2$ so that for infinitely many choices of $\epsilon$, including all irrational numbers, $h(c, X, W) = (1 + \epsilon)f_1(c) + \sum_{x \in X} f_1(x) + \sum_{\{\{w_1, w_2\}\} \in W} f_2(f_1(w_1) + f_1(w_2))$ is unique for any distinct 3-tuple of $(c, X, W)$, where $c \in \mathcal{X}$, $X \subseteq \mathcal{X}$ is a multiset with a bounded cardinality, and $W \subseteq \mathcal{W} = \{\{\{w_1, w_2\}\}|w_1, w_2 \in \mathcal{X}\}$ is a multiset of multisets with a bounded cardinality. Moreover, any function $g$ on $(c, X, W)$ can be decomposed as $g(c, X, W) = \varphi\big((1+\epsilon)f_1(c) + \sum_{x \in X} f_1(x) + \sum_{\{\{w_1, w_2\}\} \in W} f_2(f_1(w_1) + f_1(w_2))\big)$ for some function $\varphi$.*

Table 1: Comparison of expressive GNNs. $d$ is the maximum degree of nodes. $T$ is the number of triangles in the graph. $t$ is the maximum *#Message$_{NC}$* of nodes. $s$ is the maximum number of nodes in the neighborhood subgraphs, which grows exponentially with the subgraph depth. It is unknown how the expressiveness upper bound of Nested GNN compares to 3-WL

| Method | Memory | Time | Expressiveness upper bound | Scale to large graphs |
|--------|--------|------|----------------------------|-----------------------|
| GIN | $\mathcal{O}(n)$ | $\mathcal{O}(nd)$ | 1-WL | ✓ |
| 1-2-3-GNN | $\mathcal{O}(n^3)$ | $\mathcal{O}(n^4)$ | 1-WL $\sim$ 3-WL | - |
| PPGN | $\mathcal{O}(n^2)$ | $\mathcal{O}(n^3)$ | 3-WL | - |
| Nested GNN | $\mathcal{O}(ns)$ | $\mathcal{O}(nds)$ | 1-WL $\sim$ Unknown | - |
| NC-GNN (ours) | $\mathcal{O}(n + \min(m, 3T))$ | $\mathcal{O}(n(d + t))$ | 1-WL $\sim$ 3-WL | ✓ |

As Xu et al. (2019), we can use multi-layer perceptrons (MLPs) to model and learn $f_1$, $f_2$, and $\varphi$ in Lemma 5, since MLPs are universal approximators (Hornik et al., 1989; Hornik, 1991). To be specific, we use one MLP to model the compositional function $f_1^{(\ell+1)} \circ \varphi^{(\ell)}$ and another MLP to model $f_2^{(\ell)}$ for $\ell = 1, 2, \cdots, L$. At the first layer, we do not need $f_1^{(1)}$ if the input features are one-hot encodings, since there exists a function $f_2^{(1)}$ that can preserve the injectivity (See Appendix A.5 for details). Overall, one layer of our architecture can be formulated as

$$\boldsymbol{h}_v^{(\ell)} = \mathrm{MLP}_1^{(\ell)}\left(\left(1 + \epsilon^{(\ell)}\right)\boldsymbol{h}_v^{(\ell-1)} + \sum_{u \in \mathcal{N}_v} \boldsymbol{h}_u^{(\ell-1)} + \underbrace{\left(\sum_{\substack{u_1, u_2 \in \mathcal{N}_v \\ (u_1, u_2) \in E}} \mathrm{MLP}_2^{(\ell)}\left(\boldsymbol{h}_{u_1}^{(\ell-1)} + \boldsymbol{h}_{u_2}^{(\ell-1)}\right)\right)}_{\textit{The difference with GIN}}\right), \quad (6)$$

where $\epsilon^{(\ell)}$ is a learnable scalar parameter. According to Lemma 5 and Theorem 3, this simple architecture, plus an injective *readout* function, has the same expressive power as NC-1-WL. Note that this architecture follows the GIN model (Xu et al., 2019) closely. The fundamental difference between our model and GIN is highlighted in Eq. (6), which is also our key contribution. Note that if there does not exist any edges among neighbors for all nodes in a graph, the third term in Eq. (6) will be zero for all nodes, and the model will reduce to the GIN model.

**Complexity**. Suppose a graph has $n$ nodes and $m$ edges. Message passing GNNs, such as GIN, require $\mathcal{O}(n)$ memory and have $\mathcal{O}(nd)$ time complexity, where $d$ is the maximum degree of nodes. For each node, we define *#Message$_{NC}$* as the number of edges existing among neighbors of the node. An NC-GNN model as Eq (6) has $\mathcal{O}(n(d + t))$ time complexity, where $t$ denotes the maximum *#Message$_{NC}$* of nodes. Suppose there are totally $T$ triangles in the graph, in addition to $n$ node representations, we need to further store $3T$ representations as the input of MLP$_2$. If $3T > m$, we can alternatively store $(\boldsymbol{h}_{u_1} + \boldsymbol{h}_{u_2})$ for each edge $(u_1, u_2) \in E$. Thus, the memory complexity is $\mathcal{O}(n + \min(m, 3T))$. Hence, compared to message passing GNNs, our NC-GNN has a bounded memory increasement and preserves the linear time complexity with a constant factor. We compare with several existing expressive GNNs in Table 1. Our NC-GNN has much better scalability. More discussions with related works are in Section 5.

## 5 RELATED WORK

The most straightforward idea to enhance the expressiveness of message passing GNNs is to mimic the $k$-WL ($k \geq 3$) algorithms (Morris et al., 2019; 2020b; Maron et al., 2019; Chen et al., 2019). For example, Morris et al. (2019) proposes 1-2-3-GNN according to the set-based 3-WL, which is more powerful than 1-WL and less powerful than the tuple-based 3-WL. It requires $\mathcal{O}(n^3)$ memory since representations corresponding to all sets of 3 nodes needs to be stored. Moreover, without considering the sparsity of the graph, it has $\mathcal{O}(n^4)$ time complexity since each set aggregates messages from $n$ neighboring sets. Maron et al. (2019) develops PPGN based on high order invariant GNNs (Maron et al., 2018) and 2-FWL, which has the same power as 3-WL. Thereby, PPGN achieves the same power as 3-WL with $\mathcal{O}(n^2)$ memory and $\mathcal{O}(n^3)$ time complexity. Nonetheless, the computational and memory cost of these expressive models are still too high to scale to large graphs.

Another line of research for improving GNNs is to exploit subgraph information (Frasca et al., 2022). Bodnar et al. (2021b;a) perform message passing on high-order substructures of graphs, such

as simplicial and cellular complexes. Its preprocessing and message passing step are computationally expensive. Also, domain knowledge is usually required to predefine the substructure bank, while it is often unavailable in general tasks. GraphSNN (Wijesinghe & Wang, 2022) defines the overlaps between the subgraphs of each node and its neighbors, and then incorporates such overlap information into message passing scheme by using handcrafted structural coefficients. ESAN (Bevilacqua et al., 2022) employs an equivariant framework to learn from a bag of subgraphs of the graph and further proposes a subgraph selection strategy to reduce the high computational cost. Zeng et al. (2021); Sandfelder et al. (2021); Zhang & Li (2021); Zhao et al. (2022) apply GNNs to the neighborhood subgraph of each node. For example, Nested GNN (Zhang & Li, 2021) first applies a base GNN to encode the neighborhood subgraph information of each node and then employs another GNN on the subgraph-encoded representations. Since message passing is performed on $n$ neighborhood subgraphs, the memory complexity is $\mathcal{O}(ns)$ and the time complexity is $\mathcal{O}(nds)$, where $s$ is the maximum number of nodes in a neighborhood subgraph. Note that $s$ grows exponentially with the depths of the neighborhood subgraph, thus limiting the scalability. The recently proposed KP-GNN (Feng et al., 2022) focuses on formulating the K-hop message passing framework and analyzing its expressive power. In contrast, we dedicate to the consideration of edges among neighbors, leading to the powerful and efficient NC-1-WL and NC-GNN with a different proof of expressivity.

Due to the high memory and time complexity, most of the above methods are usually evaluated on graph-level tasks over small graphs, such as molecular graphs, and can be hardly applied to large graphs like social networks. Compared to these works, our approach differs fundamentally by proposing a *general* (*i.e.*, without *ad hoc* features) and *provably powerful* GNN framework, while preserving the *scalability* in terms of computational time and memory requirement. We compare our NC-GNN with several existing expressive GNNs in Table 1.

There are several other heuristic methods proposed to strengthen GNNs by adding identity-aware information (Murphy et al., 2019; Vignac et al., 2020; You et al., 2021), random features (Abboud et al., 2021; Dasoulas et al., 2021; Sato et al., 2021), predefined structural features (Li et al., 2020; Bouritsas et al., 2022) to nodes, or randomly drop node (Papp et al., 2021). Another direction is to improve GNNs in terms of the generalization ability (Puny et al., 2020). These works improve GNNs from perspectives orthogonal to ours, and thus could be used as techniques to further augment our NC-GNN. In addition, PNA (Corso et al., 2020) applies multiple aggregators to enhance the GNN performance. Most recently, Morris et al. (2022) proposes a new hierarchy, which is more fine-grained than the WL hierarchy, for graph isomorphism problem. For deeper understanding of expressive GNNs, we recommend referring to the recent surveys (Sato, 2020; Morris et al., 2021; Jegelka, 2022).

# 6 EXPERIMENTS

In this section, we perform extensive experiments to evaluate the effectiveness of the proposed NC-GNN model on real benchmarks. In particular, we consider widely used datasets from TU-Datasets (Morris et al., 2020a), Open Graph Benchmark (OGB) (Hu et al., 2020), and GNN Benchmark (Dwivedi et al., 2020). These datasets are from various domains and cover different tasks over graphs, including graph classification and node classification. Thus, they can provide a comprehensive evaluation of our method. Note that certain datasets, such as REDDIT-BINARY and ogbg-molhiv, do not have many edges among neighbors (*i.e.*, Avg. *#Message_NC* < 0.2). In this case, our NC-GNN model will almost reduce to the GIN model and thus perform nearly the same as GIN. Hence, we omit such datasets in our results. All the used datasets and their statistics, including Avg. *#Message_NC*, are summarized in Table 7, Appendix B.1. Our implementation is based on the PyG library (Fey & Lenssen, 2019). The detailed model configurations and training hyperparameters of NC-GNN on each dataset are summarized in Table 8, Appendix B.2.

**Baselines**. As shown in Eq. (6), the fundamental difference between our NC-GNN model and GIN is that we further consider modeling edges among neighbors, as highlighted in Eq. (6). Hence, comparing to GIN can directly demonstrate the effectiveness of including such information in our NC-GNN, which is the core contribution of our theoretical result. Therefore, in the following experimental results, we highlight our results if they are better than GIN, and analyze the improvements over GIN. We also consider the WL subtree kernel (Shervashidze et al., 2011) and several typical message passing GNNs as baseline; those are DCNN (Atwood & Towsley, 2016), DGCNN (Zhang et al.,

Table 2: Results (%) on TUDatasets. The top three results on each dataset are highlighted as **first**, **second**, and **third**. We also highlight the cells of NC-GNN results if they are better than GIN.

| Dataset | WL subtree | DCNN | DGCNN | GIN | PPGN | SIN | CIN | GNN-AK | GraphSNN | NC-GNN (ours) |
|---|---|---|---|---|---|---|---|---|---|---|
| IMDB-B | $73.8_{\pm 3.9}$ | $49.1_{\pm 1.4}$ | $70.0_{\pm 0.9}$ | $75.1_{\pm 5.1}$ | $73.0_{\pm 5.8}$ | $75.6_{\pm 3.2}$ | $75.6_{\pm 3.7}$ | $75.0_{\pm 4.2}$ | $77.9_{\pm 3.6}$ | $75.2_{\pm 4.5}$ |
| IMDB-M | $50.9_{\pm 3.8}$ | $33.5_{\pm 1.4}$ | $47.8_{\pm 0.9}$ | $52.3_{\pm 2.8}$ | $50.5_{\pm 3.6}$ | $52.5_{\pm 3.0}$ | $52.7_{\pm 3.1}$ | N/A | N/A | $52.5_{\pm 3.2}$ |
| COLLAB | $78.9_{\pm 1.9}$ | $52.1_{\pm 0.7}$ | $73.8_{\pm 0.5}$ | $80.2_{\pm 1.9}$ | $81.4_{\pm 1.4}$ | N/A | N/A | N/A | N/A | $82.5_{\pm 1.2}$ |
| PROTEINS | $75.0_{\pm 3.1}$ | $61.3_{\pm 1.6}$ | $75.5_{\pm 0.9}$ | $76.2_{\pm 2.8}$ | $77.2_{\pm 4.7}$ | $76.5_{\pm 3.4}$ | $77.0_{\pm 4.3}$ | $77.1_{\pm 5.7}$ | $76.8_{\pm 2.5}$ | $76.5_{\pm 4.4}$ |

2018), GCN (Kipf & Welling, 2017), and GraphSAGE (Hamilton et al., 2017). In addition, we further include the following recent methods that improve GNN expressiveness as baselines. Specifically, SIN (Bodnar et al., 2021b), CIN (Bodnar et al., 2021a), GNN-AK (Zhao et al., 2022), GraphSNN (Wijesinghe & Wang, 2022) improve the expressive power of GNNs by using the subgraph information. RingGNN (Chen et al., 2019) and PPGN (Maron et al., 2019) are models based on 3-WL.

**TUDatasets**. Following GIN (Xu et al., 2019), we first conduct experiments on four graph classification datasets from TUDatasets (Morris et al., 2020a); those are IMDB-BINARY, IMDB-MULTI, COLLAB, and PROTEINS. Note that we omit other datasets used by GIN since they do not have many edges among neighbors. We employ the same number of layers as GIN. We report the 10-fold cross validation accuracy following the protocol as (Xu et al., 2019) for fair comparison. The results of baselines are directly obtained from the literature.

As presented in Table 2, we can observe that our NC-GNN outperforms GIN on all datasets consistently. Moreover, NC-GNN performs competitively with other methods aiming at improving the GNN expressiveness. The consistent improvements over GIN can show that modeling edges among neighbors in NC-GNN is practically effective. Notably, NC-GNN achieves an obvious improvement margin of 2.3 on COLLAB. This is intuitively reasonable since the Avg. $\#Message_{NC}$ on COLLAB is much larger than other datasets, as provided in Table 7, Appendix B.1. In this case, NC-GNN can use such informative edges existing among neighbors to boost the performance.

**Open Graph Benchmark**. We also perform experiments on the large-scale dataset ogbg-ppa (Hu et al., 2020), which has over 150K graphs and is known as a more convincing testbed. The graphs in ogbg-ppa are extracted from the protein-protein association networks of different species. It formulates a graph classification task and the data are split based on species. Differing from TUDatasets, the graphs in ogbg-ppa have edge features representing the type of protein-protein association. In order to apply NC-GNN to these graphs, we further define a variant of our NC-GNN by incorporating edge features into the NC-GNN framework, inspired by the GIN model with edge features introduced by Hu et al. (2019). The details of the resulting model is given in Appendix B.3.

Since we have one more MLP than GIN at each layer, one may wonder *if our improvements are brought by the larger number of learnable parameters, instead of our claimed expressiveness.* Thus, here we compare with GIN under the same parameter budget. Specifically, we use the same number of layers as GIN to ensure the same receptive field, and tune the hidden dimension to obtain an NC-GNN model that has the similar number of learnable parameters as GIN. Following (Hu et al., 2020), we compare the best validation accuracy and the test accuracy at the best validation epoch. We also include the training accuracy at the best validation epoch for reference. Results over 10 random runs are reported. The results of GIN are obtained from the official benchmark leaderboard.

As reported in Table 3, our NC-GNN consistently achieves better results in terms of validation accuracy and test accuracy. Specifically, our NC-GNN model outperforms GIN on the test set by an obvious absolute margin of 3.02. Note that the only difference between

Table 3: Results (%) on ogbg-ppa. All models in this table consider edge features. We highlight the cells of NC-GNN results if they are better than GIN.

| Model | # Param | Training Acc. | Validation Acc. | Test Acc. |
|---|---|---|---|---|
| GIN | 1836942 | $97.55_{\pm 0.52}$ | $65.62_{\pm 1.07}$ | $68.92_{\pm 1.00}$ |
| NC-GNN (ours) | 1754445 | $99.37_{\pm 0.16}$ | $66.82_{\pm 0.67}$ | $71.94_{\pm 0.43}$ |

NC-GNN and GIN is that edges among neighbors are modeled and considered in NC-GNN. Thus, the obvious improvements over GIN can demonstrate the practical effectiveness of incorporating such information. Therefore, combining with the previous experiments on TUDatasets, we can conclude that our NC-GNN not only has theoretically provable expressiveness, but also achieves good empirical performance on real-world tasks.

**GNN Benchmark**. In addition to graph classification tasks, we further experiment with node classification tasks on two datasets, PATTERN and CLUSTER, from GNN Benchmark (Dwivedi et al., 2020). PATTERN and CLUSTER respectively contain 14K and 12K graphs generated from Stochastic Block Models (Abbe, 2017), a widely used mathematical modeling method for studying communities in social networks. The tasks on these two datasets it to classify nodes in each graph. To be specific, on PATTERN, the goal is to determine if a node belongs to specific predetermined subgraph patterns. On CLUSTER, we aim at categorizing each node to its belonging community. The details of the construction of these datasets are available in (Dwivedi et al., 2020).

We compare with typical message passing GNNs, including GIN, and two methods that mimic 3-WL; those are PPGN and RingGNN. To ensure fair comparison, we follow Dwivedi et al. (2020) to compare different methods under two budgets of learnable parameters, 100K and 500K, by tuning the number of layers and the hidden dimensions. Average results over 4 random runs are reported in Table 4, where the results of baselines are obtained from (Dwivedi et al., 2020). On each dataset, we also present the absolute improvement margin of our NC-GNN over GIN, denoted as $\Delta^{\uparrow}$, by comparing their corresponding best result.

We observe that our NC-GNN obtains significant improvements over GIN. To be specific, NC-GNN remarkably outperforms GIN by an absolute margin of $1.142$ and $12.002$ on PATTERN and CLUSTER, respectively. This further strongly demonstrates the effectiveness of modeling the information of edges among neighbors, which aligns with our theoretical results. Notably, NC-GNN obtains outstanding performance on CLUSTER. Since the task on CLUSTER is to identify communities, we reasonably conjecture that considering which neighbors are connected is essential for inferring communities. Thus, we believe that our NC-GNN can be a strong basic method for tasks over social network graphs.

Table 4: Results (%) on GNN Benchmark. The top three results on each dataset are highlighted as **first**, **second**, and **third**. We also highlight the cells of NC-GNN results if they are better than GIN.

| Model | # Layers | PATTERN | | CLUSTER | |
|---|---|---|---|---|---|
| | | # Param | Test Acc. | # Param | Test Acc. |
| GCN | 4 | 100923 | $85.498_{\pm0.045}$ | 101655 | $47.828_{\pm1.510}$ |
| | 16 | 500823 | $85.614_{\pm0.032}$ | 501687 | $69.026_{\pm1.372}$ |
| GraphSAGE | 4 | 101739 | $50.516_{\pm0.001}$ | 102187 | $50.454_{\pm0.145}$ |
| | 16 | 502842 | $50.492_{\pm0.001}$ | 503350 | $63.844_{\pm0.110}$ |
| GIN | 4 | 100884 | $85.590_{\pm0.011}$ | 103544 | $58.384_{\pm0.236}$ |
| | 16 | 508574 | $85.387_{\pm0.136}$ | 517570 | $64.716_{\pm1.553}$ |
| RingGNN | 2 | 105206 | $86.245_{\pm0.013}$ | 104746 | $42.418_{\pm20.063}$ |
| | 2 | 504766 | $86.244_{\pm0.025}$ | 524202 | $22.340_{\pm0.000}$ |
| | 8 | 505749 | Diverged | 514380 | Diverged |
| PPGN | 3 | 103572 | $85.661_{\pm0.353}$ | 105552 | $57.130_{\pm6.539}$ |
| | 3 | 502872 | $85.341_{\pm0.207}$ | 507252 | $55.489_{\pm7.863}$ |
| | 8 | 581716 | Diverged | 586788 | Diverged |
| NC-GNN (ours) | 4 | 106756 | $86.627_{\pm0.017}$ | 107320 | $69.335_{\pm0.357}$ |
| | 4 | 506512 | $86.732_{\pm0.007}$ | 508428 | $69.838_{\pm0.135}$ |
| | 16 | 506512 | $86.607_{\pm0.119}$ | 508428 | $76.718_{\pm0.071}$ |
| $\Delta^{\uparrow}$ (over GIN) | | | $1.142 \uparrow$ | | $12.002 \uparrow$ |

Moreover, NC-GNN achieves much better empirical performance than RingGNN and PPGN, although they theoretically mimic 3-WL. It is observed that these 3-WL based methods are difficult to train and thus having fluctuating performance (Dwivedi et al., 2020). In contrast, our NC-GNN is easier to train since it preserves the locality of message passing, thereby being more practically effective.

**Time analysis**. In Table 5, we compare the real training time of models with 100K learnable parameters on IMDB-B, PATTERN, and CLUSTER. We can observe that our NC-GNN is much more efficient than PPGN, since our NC-GNN preserves the linear time complexity *w.r.t.* number of nodes as GIN, according to the analysis in Table 1. Compared to GIN, the increasement of the real running time of our NC-GNN depends on the number of edges among neighbors. For example, the time consumption of NC-GNN is similar to GIN on IMDB-B, since the Avg. *#Message$_{NC}$* is considerably smaller than that in PATTERN and CLUSTER. Overall, our NC-GNN is shown to be more powerful than GIN theoretically and empirically, while maintaining the scalability with reasonable overhead.

Table 5: Comparison of real training time.

| dataset | IMDB-B | | PATTERN | | CLUSTER | |
|---|---|---|---|---|---|---|
| Avg. *#Message$_{NC}$* | 59.5 | | 3440.1 | | 1301.5 | |
| Model | Acc. | Time/epoch | Acc. | Time/epoch | Acc. | Time/epoch |
| GIN | $75.1_{\pm5.1}$ | 1.075s | $85.590_{\pm0.011}$ | 7.431s | $58.384_{\pm0.236}$ | 6.180s |
| PPGN | $73.0_{\pm5.8}$ | 4.586s | $85.661_{\pm0.353}$ | 147.063s | $57.130_{\pm6.539}$ | 144.106s |
| NC-GNN (ours) | $75.2_{\pm4.5}$ | 1.252s | $86.627_{\pm0.017}$ | 44.882s | $69.838_{\pm0.135}$ | 28.505s |

**Thorough comparison with subgraph GNNs**. We further perform a comprehensive empirical comparison with subgraph GNNs. Specifically, we compare to GIN-AK$^+$ (Zhao et al., 2022), a representative method in the subgraph GNN family, on test accuracy, training time per epoch, total training time for convergence, GPU memory usage, MACS, and inference time. For each experiment,

Table 6: The thorough comparison between NCGNN and GIN-AK$^+$ on PATTERN and CLUSTER.

| Datasets | Methods | # Layers | # Param | Test Acc.$^\uparrow$ | Time/epoch$\downarrow$ | Total Time$\downarrow$ | GPU Memory$\downarrow$ | MACS$\downarrow$ | Inference Time$\downarrow$ |
|---|---|---|---|---|---|---|---|---|---|
| PATTERN | GIN-AK$^+$ | 4 | 601134 | **86.836**$_{\pm0.007}$ | 77.503s | 0.667h | 31434MB | 176.445G | 32.188s |
| | NC-GNN (ours) | 4 | 552096 | 86.717$_{\pm0.069}$ | **42.517**s | **0.555**h | **15625**MB | **1.142**G | **12.025**s |
| CLUSTER | GIN-AK$^+$ | 16 | 602586 | 76.502$_{\pm0.210}$ | 148.983s | 1.420h | 32142MB | 110.518G | 27.163s |
| | NC-GNN (ours) | 16 | 562948 | **76.992**$_{\pm0.063}$ | **48.869**s | **0.679**h | **22386**MB | **0.841**G | **5.1763**s |

we run it four times and report the average results for test accuracy, training time per epoch, total time consumed to achieve the best epoch, and GPU memory consumption while keeping the same batch size. In order to compare the computational cost, FLOPS is commonly used as the number of floating operations for the model(Tan & Le, 2019). Here we use a similar metric MACS to calculate the average multiply-accumulate operations for each graph in the test set. Note that each multiply-accumulate operation includes two float operations. The results are summarized in Table 6. Our NC-GNN achieves competitive accuracies as GIN-AK$^+$, while being more efficient in training, including training time per epoch and total training time. In addition, we use less GPU memory since we do not have to consider updating node representations for all the nodes in the expanded subgraphs as GIN-AK$^+$. More importantly, the MACS overhead of GIN-AK$^+$ is 100x more than our NC-GNN. Since our NC-GNN calculates each node representation from the original graph instead of the expanded subgraphs, it can save huge MACS overhead during the forward procedure. To further show the advantage of fewer MACS overhead, we provide the inference time comparison and our NC-GNN takes less time during inference. Overall, NC-GNN reaches a sweet spot between expressivity and scalability, from both theoretical and practical observations.

# 7 CONCLUSIONS AND OUTLOOKS

In this work, based on our proposed NC-1-WL, we present NC-GNN, a general, provably powerful, and scalable framework for graph representation learning. In addition to the theoretical expressiveness, we empirically demonstrate that NC-GNN achieves outstanding performance on various real benchmarks. Thus, we anticipate that NC-GNN will become an important base model for learning from graphs, especially social network graphs. To further improve the expressiveness of NC-GNN, in future work, we may consider how to effectively and efficiently model the interactions between two edges that exist among neighbors.

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

# A PROOFS OF THEOREMS AND LEMMAS

## A.1 PROOF OF THEOREM 1

**Theorem 1.** *NC-1-WL is strictly more powerful than 1-WL in distinguishing non-isomorphic graphs.*

*Proof.* To prove that NC-1-WL is strictly more powerful than 1-WL, we prove the correctness of the following two statements. (1) If two graphs are determined to be isomorphic by NC-1-WL, then they must be indistinguishable by 1-WL as well. (2) There exist at least two non-isomorphic graphs that cannot be distinguished by 1-WL but can be distinguished by NC-1-WL.

(1) Assume two graphs $\mathcal{G} = (V, E, \boldsymbol{X})$ and $\mathcal{H} = (P, F, \boldsymbol{Y})$ cannot be distinguished by NC-1-WL. Then, according to Algorithm 1, at any iteration $\ell = 1, 2, \cdots$, we have $\left\{\left\{\left(c_v^{(\ell-1)}, \{\{c_u^{(\ell-1)}|u \in \mathcal{N}_v\}\}, \{\{\{\{c_{u_1}^{(\ell-1)}, c_{u_2}^{(\ell-1)}\}\}|u_1, u_2 \in \mathcal{N}_v, (u_1, u_2) \in E\}\}\right)|v \in V\right\}\right\} = \left\{\left\{\left(d_p^{(\ell-1)}, \{\{d_q^{(\ell-1)}|q \in \mathcal{N}_p\}\}, \{\{\{\{d_{q_1}^{(\ell-1)}, d_{q_2}^{(\ell-1)}\}\}|q_1, q_2 \in \mathcal{N}_p, (q_1, q_2) \in F\}\}\right)|p \in P\right\}\right\}$. This naturally implies that, at any iteration $\ell = 1, 2, \cdots$, we have $\left\{\left\{\left(c_v^{(\ell-1)}, \{\{c_u^{(\ell-1)}|u \in \mathcal{N}_v\}\}\right)|v \in V\right\}\right\} = \left\{\left\{\left(d_p^{(\ell-1)}, \{\{d_q^{(\ell-1)}|q \in \mathcal{N}_p\}\}\right)|p \in P\right\}\right\}$. This indicates that 1-WL cannot distinguish graph $\mathcal{G}$ and graph $\mathcal{H}$ as well.

(2) In Figure 1 (a), we provide several pairs as example to show that there exist such non-isomorphic graphs that can be distinguished by NC-1-WL but cannot be distinguished by 1-WL. ☐

## A.2 PROOF OF THEOREM 2

**Theorem 2.** *NC-1-WL is strictly less powerful than 3-WL in distinguishing non-isomorphic graphs.*

*Proof.* To prove that NC-1-WL is strictly less powerful than 3-WL, we prove the correctness of the following two statements. (1) If two graphs are determined to be isomorphic by 3-WL, then they must be indistinguishable by NC-1-WL as well. (2) There exist at least two non-isomorphic graphs that cannot be distinguished by NC-1-WL but can be distinguished by 3-WL.

We first describe the details of $k$-WL in Algorithm 2, following Sato (2020). $k$-WL aims at coloring each $k$-tuple of nodes, denoted as $\boldsymbol{v} \in V^k$. The $i$-th neighborhood of each tuple $\boldsymbol{v} = (v_1, v_2, \cdots, v_k)$ is defined as $\mathcal{N}_{\boldsymbol{v},i} = \{(v_1, \cdots, v_{i-1}, s, v_{i+1}, \cdots, v_k)|s \in V\}$. Similarly, The $i$-th neighborhood of each tuple $\boldsymbol{p} = (p_1, p_2, \cdots, p_k)$ is defined as $\mathcal{N}_{\boldsymbol{p},i} = \{(p_1, \cdots, p_{i-1}, t, p_{i+1}, \cdots, p_k)|t \in P\}$. The initial color of each tuple $\boldsymbol{v}$ is determined by the isomorphic type of the subgraph induced by the tuple, *i.e.*, $\mathcal{G}[\boldsymbol{v}]$. (See Sato (2020) for details). Note that the nodes in $\mathcal{G}[\boldsymbol{v}]$ are ordered based on their orders in the tuple $\boldsymbol{v}$. Thus, $\text{HASH}(\mathcal{G}[\boldsymbol{v}]) = \text{HASH}(\mathcal{H}[\boldsymbol{p}])$ iff (a) $\boldsymbol{x}_{v_i} = \boldsymbol{y}_{p_i}$ for $i = 1, 2, \cdots, k$ and (b) $(v_i, v_j) \in E$ iff $(p_i, p_j) \in F$.

(1) Assume two graphs $\mathcal{G} = (V, E, \boldsymbol{X})$ and $\mathcal{H} = (P, F, \boldsymbol{Y})$ are determined to be isomorphic by 3-WL. $\mathcal{G}$ and $\mathcal{H}$ have the same number of nodes[3], denoted as $n$. Then, according to Algorithm 2, we have $\{\{c_{\boldsymbol{v}}^{(0)}|\boldsymbol{v} \in V^k\}\} = \{\{d_{\boldsymbol{p}}^{(0)}|\boldsymbol{p} \in P^k\}\}$. There always exists an injective mapping $g : V \to P$ such that $c_{\boldsymbol{v}}^{(0)} = d_{g(\boldsymbol{v})}^{(0)}$ (*i.e.*, $\text{HASH}(\mathcal{G}[\boldsymbol{v}]) = \text{HASH}(\mathcal{H}[g(\boldsymbol{v})])$), $\forall \boldsymbol{v} \in V^k$. Here, we directly apply $g$ to a tuple $\boldsymbol{v}$ for ease of notation, which means $g(\boldsymbol{v}) = g((v_1, v_2, v_3)) = (g(v_1), g(v_2), g(v_3))$. Without losing generality, we assume $g$ maps $v_j$ to $p_j$ for $j = 1, 2, \cdots, n$. Then, we can obtain the following results.

(a) We can consider the tuples $\boldsymbol{v} = (v_j, v_j, v_j), \forall v_j \in V$. Given $c_{\boldsymbol{v}}^{(0)} = d_{g(\boldsymbol{v})}^{(0)}$, we can derive $\boldsymbol{x}_{v_j} = \boldsymbol{y}_{p_j}, \forall v_j \in V$.

(b) We further consider the tuples $\boldsymbol{v} = (v_j, v_j, v_r), \forall v_j \in V$. According to $c_{\boldsymbol{v}}^{(0)} = d_{g(\boldsymbol{v})}^{(0)}$, we have $\boldsymbol{x}_{v_r} = \boldsymbol{y}_{p_r}$. We can also have $p_r \in \mathcal{N}_{p_j}$ iff $v_r \in \mathcal{N}_{v_j}$ (Otherwise, $c_{\boldsymbol{v}}^{(0)} \neq d_{g(\boldsymbol{v})}^{(0)}$).

---

[3]Two graphs with different numbers of nodes can be easily distinguished by comparing the multisets of node colors, given that the cardinalities of the two multisets are different.

---

**Algorithm 2** $k$-WL for graph isomorphism test

---

**Input:** Two graphs $\mathcal{G} = (V, E, \boldsymbol{X})$ and $\mathcal{H} = (P, F, \boldsymbol{Y})$
$c_{\boldsymbol{v}}^{(0)} \leftarrow \text{HASH}(\mathcal{G}[\boldsymbol{v}]), \forall \boldsymbol{v} \in V^k$
$d_{\boldsymbol{p}}^{(0)} \leftarrow \text{HASH}(\mathcal{H}[\boldsymbol{p}]), \forall \boldsymbol{p} \in P^k$
**repeat** $(\ell = 1, 2, \cdots)$
    **if** $\{\{c_{\boldsymbol{v}}^{(\ell-1)} | \boldsymbol{v} \in V^k\}\} \neq \{\{d_{\boldsymbol{p}}^{(\ell-1)} | \boldsymbol{v} \in P^k\}\}$ **then**
        **return** $\mathcal{G} \not\simeq \mathcal{H}$
    **end if**
    **for** $\boldsymbol{v} \in V^k$ **do**
        $c_{\boldsymbol{v},i}^{(\ell)} = \{\{c_{\boldsymbol{u}}^{(\ell-1)} | \boldsymbol{u} \in \mathcal{N}_{\boldsymbol{v},i}\}\}, \quad \text{for } i = 1, 2, \cdots, k$
        $c_{\boldsymbol{v}}^{(\ell)} \leftarrow \text{HASH}\left(c_{\boldsymbol{v}}^{(\ell-1)}, c_{\boldsymbol{v},1}^{(\ell)}, c_{\boldsymbol{v},2}^{(\ell)}, \cdots, c_{\boldsymbol{v},k}^{(\ell)}\right)$
    **end for**
    **for** $\boldsymbol{p} \in P^k$ **do**
        $d_{\boldsymbol{p},i}^{(\ell)} = \{\{d_{\boldsymbol{q}}^{(\ell-1)} | \boldsymbol{q} \in \mathcal{N}_{\boldsymbol{p},i}\}\}, \quad \text{for } i = 1, 2, \cdots, k$
        $d_{\boldsymbol{p}}^{(\ell)} \leftarrow \text{HASH}\left(d_{\boldsymbol{p}}^{(\ell-1)}, d_{\boldsymbol{p},1}^{(\ell)}, d_{\boldsymbol{p},2}^{(\ell)}, \cdots, d_{\boldsymbol{p},k}^{(\ell)}\right)$
    **end for**
**until** convergence
**return** $\mathcal{G} \simeq \mathcal{H}$

---

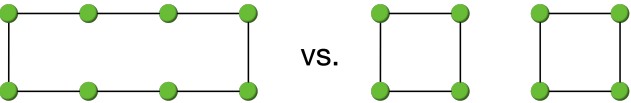

Figure 2: Two graphs, adapted from Sato (2020), that cannot be distinguished by NC-1-WL but can be distinguished by 3-WL.

(c) At last, we consider the tuples $\boldsymbol{v} = (v_j, v_r, v_w), \forall v_j \in V$. Similarly, based on $c_{\boldsymbol{v}}^{(0)} = d_{g(\boldsymbol{v})}^{(0)}$, we can obtain $\boldsymbol{x}_{v_r} = \boldsymbol{y}_{p_r}$ and $\boldsymbol{x}_{v_w} = \boldsymbol{y}_{p_w}$. Also, we can have $p_r, p_w \in \mathcal{N}_{p_j}$ iff $v_r, v_w \in \mathcal{N}_{v_j}$, and $(p_r, p_w) \in F$ iff $(v_r, v_w) \in E$.

Now, we consider performing NC-1-WL (Algorithm 1) on these two graphs $\mathcal{G} = (V, E, \boldsymbol{X})$ and $\mathcal{H} = (P, F, \boldsymbol{Y})$ to color each node $v \in V$ and $p \in P$. We have the same injective mapping $g : V \rightarrow P$ as above. Based on (a), we have $\boldsymbol{x}_v = \boldsymbol{y}_{g(v)}, \forall v \in V$, which indicates $c_v^{(0)} = d_{g(v)}^{(0)}, \forall v \in V$ in the NC-1-WL coloring process. Similarly, according to (b) and (c), we have $\{\{c_u^{(0)} | u \in \mathcal{N}_v\}\} = \{\{d_q^{(0)} | q \in \mathcal{N}_{g(v)}\}\}, \forall v \in V$ and $\{\{\{\{c_{u_1}^{(0)}, c_{u_2}^{(0)}\}\} | u_1, u_2 \in \mathcal{N}_v, (u_1, u_2) \in E\}\} = \{\{\{\{d_{q_1}^{(0)}, d_{q_2}^{(0)}\}\} | q_1, q_2 \in \mathcal{N}_{g(v)}, (q_1, q_2) \in F\}\}, \forall v \in V$, respectively. Therefore, with initial colors satisfying such conditions, NC-1-WL cannot distinguish $\mathcal{G}$ and $\mathcal{H}$ since $c_v^{(l-1)} = d_{g(v)}^{(l-1)}, \forall v \in V$ always holds for $\ell = 1, 2, \cdots$. In other words, $\{\{c_v^{(\ell-1)} | v \in V\}\} = \{\{d_p^{(\ell-1)} | p \in P\}\}$ always holds no matter how many iterations (*i.e.*, $\ell$) we apply.

(2) In Figure 2, we provide two non-isomorphic graphs that can be distinguished by 3-WL but cannot be distinguished by NC-1-WL. For these two graphs, our NC-1-WL reduces to 1-WL since there does not exist any neighbors that are communicating. □

### A.3 PROOF OF THEOREM 3

**Theorem 3.** *Let $\mathcal{M} : \mathcal{G} \rightarrow \mathbb{R}^d$ be an NC-GNN model with a sufficient number of layers following Eq. (5). $\mathcal{M}$ is as powerful as NC-1-WL in distinguishing non-isomorphic graphs if the following conditions hold: (1) At each layer $\ell$, $f_{communicate}^{(\ell)}$, $f_{aggregate}^{(\ell)}$, and $f_{update}^{(\ell)}$ are injective. (2) The final readout function $f_{readout}$ is injective.*

*Proof.* We prove the theorem by showing that an NC-GNN model that satisfies the conditions can yield different embeddings for any two graphs that are determined to be non-isomorphic by NC-1-WL. We denote such model as $\mathcal{M}$. Assume two graphs $\mathcal{G}_1 = (V_1, E_1, \boldsymbol{X}_1)$ and $\mathcal{G}_2 = (V_2, E_2, \boldsymbol{X}_2)$ are determined to be non-isomorphic by NC-1-WL at iteration $L$. Given that $f_{\text{readout}}$ of $\mathcal{M}$ can injectively map different multisets of node features into different embeddings, we only need to demonstrate that $\mathcal{M}$, with a sufficient number of layers, can map $\mathcal{G}_1$ and $\mathcal{G}_2$ to different multisets of node features.

To achieve this, following Xu et al. (2019), we show that, for any iteration $\ell$, there always exists an injective function $\varphi$ such that $\boldsymbol{h}_v^{(\ell)} = \varphi(c_v^{(\ell)})$, where $\boldsymbol{h}_v^{(\ell)}$ is the node representation given by the model $\mathcal{M}$ and $c_v^{(\ell)}$ is the color produced by NC-1-WL. We will show this by induction. Note that here $v$ represents a general node that can be a node in $\mathcal{G}_1$ or $\mathcal{G}_2$.

Let $\phi$ denote the injective hash function used in NC-1-WL. For $\ell = 0$, we have $c_v^{(0)} = \phi(\boldsymbol{x}_v)$ and $\boldsymbol{h}_v^{(0)} = \boldsymbol{x}_v$. Thus, $\varphi$ could be $\phi^{-1}$ for $\ell = 0$. Suppose there exists an injective function $\varphi$ such that $\boldsymbol{h}_v^{(\ell-1)} = \varphi(c_v^{(\ell-1)}), \forall v \in V_1 \cup V_2$, we show that there also exists such an injective function for iteration $\ell$. According to Eq. (5), we have

$$
\begin{aligned}
\boldsymbol{h}_v^{(\ell)} = f_{\text{update}}{}^{(\ell)} &\left( \boldsymbol{h}_v^{(\ell-1)}, f_{\text{aggregate}}{}^{(\ell)} \left( \{\{\boldsymbol{h}_u^{(\ell-1)} | u \in \mathcal{N}_v \}\} \right), \right. \\
&\left. f_{\text{communicate}}{}^{(\ell)} \left( \{\{\{\{\boldsymbol{h}_{u_1}^{(\ell-1)}, \boldsymbol{h}_{u_2}^{(\ell-1)}\}\}| u_1, u_2 \in \mathcal{N}_v, (u_1, u_2) \in E\}\} \right) \right).
\end{aligned}
\tag{7}
$$

According to $\boldsymbol{h}_v^{(\ell-1)} = \varphi(c_v^{(\ell-1)})$, we further have

$$
\begin{aligned}
\boldsymbol{h}_v^{(\ell)} = f_{\text{update}}{}^{(\ell)} &\left( \varphi(c_v^{(\ell-1)}), f_{\text{aggregate}}{}^{(\ell)} \left( \{\{\varphi(c_u^{(\ell-1)}) | u \in \mathcal{N}_v \}\} \right), \right. \\
&\left. f_{\text{communicate}}{}^{(\ell)} \left( \{\{\{\{\varphi(c_{u_1}^{(\ell-1)}), \varphi(c_{u_2}^{(\ell-1)})\}\}| u_1, u_2 \in \mathcal{N}_v, (u_1, u_2) \in E\}\} \right) \right),
\end{aligned}
\tag{8}
$$

where $f_{\text{communicate}}{}^{(\ell)}$, $f_{\text{aggregate}}{}^{(\ell)}$, $f_{\text{update}}{}^{(\ell)}$, and $\varphi$ are all injective functions. Since the composition of injective functions is also injective, there must exist some injective function $\psi$ such that

$$
\boldsymbol{h}_v^{(\ell)} = \psi \left( c_v^{(\ell-1)}, \{\{c_u^{(\ell-1)} | u \in \mathcal{N}_v\}\}, \{\{\{\{c_{u_1}^{(\ell-1)}, c_{u_2}^{(\ell-1)}\}\}| u_1, u_2 \in \mathcal{N}_v, (u_1, u_2) \in E\}\} \right). \tag{9}
$$

Then, we can obtain

$$
\begin{aligned}
\boldsymbol{h}_v^{(\ell)} &= \psi \left( \phi^{-1} \left( \phi \left( c_v^{(\ell-1)}, \{\{c_u^{(\ell-1)} | u \in \mathcal{N}_v\}\}, \{\{\{\{c_{u_1}^{(\ell-1)}, c_{u_2}^{(\ell-1)}\}\}| u_1, u_2 \in \mathcal{N}_v, (u_1, u_2) \in E\}\} \right) \right) \right), \\
&= \psi \left( \phi^{-1} \left( c_v^{(\ell)} \right) \right).
\end{aligned}
\tag{10}
$$

Then, $\varphi = \psi \circ \phi^{-1}$ is an injective function such that $\boldsymbol{h}_v^{(\ell)} = \varphi(c_v^{(\ell)}), \forall v \in V_1 \cup V_2$.

Therefore, it is proved that for any iteration $\ell$, there always exists an injective function $\varphi$ such that $\boldsymbol{h}_v^{(\ell)} = \varphi(c_v^{(\ell)})$. Since NC-1-WL determines $\mathcal{G}_1$ and $\mathcal{G}_2$ to be non-isomorphic at iteration $L$, we have $\{\{c_v^{(L)} | v \in V_1\}\} \neq \{\{c_v^{(L)} | v \in V_2\}\}$. As proved above, we also have $\{\{\boldsymbol{h}_v^{(L)} | v \in V_1\}\} = \{\{\varphi(c_v^{(L)}) | v \in V_1\}\}$, $\{\{\boldsymbol{h}_v^{(L)} | v \in V_2\}\} = \{\{\varphi(c_v^{(L)}) | v \in V_2\}\}$, and $\varphi$ is injective. Hence, the multisets of node features produced by $\mathcal{M}$ for $\mathcal{G}_1$ and $\mathcal{G}_2$ are also different, *i.e.*, $\{\{\boldsymbol{h}_v^{(L)} | v \in V_1\}\} \neq \{\{\boldsymbol{h}_v^{(L)} | v \in V_2\}\}$, which indicates that the NC-GNN model $\mathcal{M}$ can also distinguish $\mathcal{G}_1$ and $\mathcal{G}_2$. $\square$

### A.4    PROOF OF LEMMA 4

**Lemma 4.** *Assume $\mathcal{X}$ is countable. There exist two functions $f_1$ and $f_2$ so that $h(X, W) = \sum_{x \in X} f_1(x) + \sum_{\{\{w_1, w_2\}\} \in W} f_2(f_1(w_1) + f_1(w_2))$ is unique for any distinct pair of $(X, W)$, where $X \subseteq \mathcal{X}$ is a multiset with a bounded cardinality and $W \subseteq \mathcal{W} = \{\{\{w_1, w_2\}\} | w_1, w_2 \in \mathcal{X}\}$ is a multiset of multisets with a bounded cardinality. Moreover, any function $g$ on $(X, W)$ can be decomposed as $g(X, W) = \phi\big(\sum_{x \in X} f_1(x) + \sum_{\{\{w_1, w_2\}\} \in W} f_2(f_1(w_1) + f_1(w_2))\big)$ for some function $\phi$.*

*Proof.* To prove this Lemma, we need the following fact, which is also used by Xu et al. (2019).

**Fact 1**. Assume $\mathcal{X}$ is countable. $h(X) = \sum_{x \in X} N^{-Z(x)}$ is unique for any multiset $X \subseteq \mathcal{X}$ of bounded cardinality, where the mapping $Z : \mathcal{X} \to \mathbb{N}$ is an injection from $x \in \mathcal{X}$ to natural numbers and $N \in \mathbb{N}$ satisfies $N > |X|$ for all $X$.

To prove the correctness of this fact, we show that $X$ can be uniquely obtained from the value of $h(X)$. Following the notations in our main texts, we formally denote $X = (S_X, m_X)$, where $S_X$ is the underlying set of $X$ and $m_X : S_X \to \mathbb{Z}^+$ gives the multiplicity of the elements in $S_X$. Hence, we need to uniquely determine the elements in $S_X$ and their corresponding multiplicities, using the value of $h(X)$. Let $\{x_1, x_2, \cdots, x_n\}$ denote the countable set $\mathcal{X}$ ($n$ could go infinitely). Without losing generality, we assume $Z$ maps $x_1 \to 0$, $x_2 \to 1$, *etc.*. Then we can compute $(q, r) = h(X) \ divmod \ N^{-0}$, where $q$ is the quotient and $r$ is the remainder. If $q = 0$, we can conclude $x_1$ is not in $S_X$. If $q > 0$, then $x_1$ is in $S_X$ and $q$ gives the multiplicity of $x_1$. Afterwards, we use the remainder $r$ to replace $h(X)$ and compute $(q, r) = h(X) \ divmod \ N^{-1}$, and the results can be used to infer if $x_2$ is in $S_X$ and its multiplicity. We can do this recursively until $r = 0$. Note that $X$ has a bounded cardinality and $N \in \mathbb{N}$ satisfies $N > |X|$ for all $X$. Otherwise, Fact 1 will not hold. Here we provide an example to show the correctness of Fact 1. Let a multiset $X = \{\{x_1, x_3, x_3\}\}$ and $Z$ injectively maps the elements in $X$ to natural numbers, thus obtaining a multiset $\{\{0, 2, 2\}\}$. Let $N = 4$. We have $h(X) = \sum_{x \in X} N^{-Z(x)} = 4^{-0} + 4^{-2} + 4^{-2} = 9/8$. Next, following our description above, we show how we can infer $X$ by the value of $h(X)$. First, according to $9/8 \ divmod \ 4^{-0} = (1, 1/8)$, we can conclude that there is one $x_1$ in $X$. Then, with $1/8 \ divmod \ 4^{-1} = (0, 1/8)$, we can infer that $x_2$ is not in $X$. Finally, we have $1/8 \ divmod \ 4^{-2} = (2, 0)$, which indicates that there are two $x_3$ in $X$. We can stop the process since the remainder reaches 0.

Let us go back to the proof of Lemma 4. Since $\mathcal{X}$ is countable, $\mathcal{W} = \{\{\{w_1, w_2\}\}|w_1, w_2 \in \mathcal{X}\}$ is also countable. Because both $X$ and $W$ have bounded cardinalities, we can find an number $N \in \mathbb{N}$ such that $N > max(|X| + |W|, 2)$ for all $(X, W)$ pairs. Let $Z_1 : \mathcal{X} \to \mathbb{N}_{odd}$ be an injection from $x \in \mathcal{X}$ to odd natural numbers. We consider $f_1 = N^{-Z_1(x)}$. For ease of notation, we let $\psi(\{\{w_1, w_2\}\}) = f_1(w_1) + f_1(w_2)$. According to Fact 1, $\psi(\{\{w_1, w_2\}\})$ is unique for any $\{\{w_1, w_2\}\} \in \mathcal{W}$. We define the set $\mathcal{Y} = \{\psi(\{\{w_1, w_2\}\})|w_1, w_2 \in \mathcal{X}\}$. Thus, $\mathcal{Y}$ is also countable as $\mathcal{W}$. We consider $Z_2 : \mathcal{Y} \to \mathbb{N}_{even}$ be an injection from $y \in \mathcal{Y}$ to even natural numbers and $f_2 = N^{-Z_2(y)}$. Then, the resulting $h(X, W) = \sum_{x \in X} f_1(x) + \sum_{\{\{w_1, w_2\}\} \in W} f_2(f_1(w_1) + f_1(w_2))$ is an injective function on $(X, W)$. In other words, we can uniquely determine $(X, W)$ from the value of $h(X, W)$. To be specific, from the value of $h(X, W)$, we can infer the histograms of natural numbers as we show in Fact 1. Then, we can uniquely obtain $X$ (*i.e.*, its underlying set $S_x$ and multiplicities.) based on the odd natural numbers. According to the even natural numbers, we can infer $\{\{\psi(\{\{w_1, w_2\}\})|\{\{w_1, w_2\}\} \in W\}\}$. Further, since $\psi(\{\{w_1, w_2\}\})$ is injective, we can uniquely obtain $W$.

For any function $g$ on $(X, W)$, we can construct a function $\phi$ such that $\phi\big(h(X, W)\big) = g(X, W)$. This is always achievable since $h(X, W)$ is injective. $\qquad\square$

### A.5 Proof of Lemma 5

**Lemma 5.** *Assume $\mathcal{X}$ is countable. There exist two functions $f_1$ and $f_2$ so that for infinitely many choices of $\epsilon$, including all irrational numbers, $h(c, X, W) = (1 + \epsilon)f_1(c) + \sum_{x \in X} f_1(x) + \sum_{\{\{w_1, w_2\}\} \in W} f_2(f_1(w_1) + f_1(w_2))$ is unique for any distinct 3-tuple of $(c, X, W)$, where $c \in \mathcal{X}$, $X \subseteq \mathcal{X}$ is a multiset with a bounded cardinality, and $W \subseteq \mathcal{W} = \{\{\{w_1, w_2\}\}|w_1, w_2 \in \mathcal{X}\}$ is a multiset of multisets with a bounded cardinality. Moreover, any function $g$ on $(c, X, W)$ can be decomposed as $g(c, X, W) = \varphi\big((1 + \epsilon)f_1(c) + \sum_{x \in X} f_1(x) + \sum_{\{\{w_1, w_2\}\} \in W} f_2(f_1(w_1) + f_1(w_2))\big)$ for some function $\varphi$.*

*Proof.* We consider the same functions $f_1 = N^{-Z_1(x)}$ and $f_2 = N^{-Z_2(y)}$ as in our proof for Lemma 4. We prove this lemma by showing that, if $\epsilon$ is an irrational number, $h(c, X, W) \neq h(c', X', W')$ holds for any $(c, X, W) \neq (c', X', W')$. We need to consider the following two cases.

(1) If $c = c'$ but $(X, W) \neq (X', W')$, according to Lemma 4, we have $\sum_{x \in X} f_1(x) + \sum_{\{\{w_1, w_2\}\} \in W} f_2(f_1(w_1) + f_1(w_2)) \neq \sum_{x \in X'} f_1(x) + \sum_{\{\{w_1, w_2\}\} \in W'} f_2(f_1(w_1) + f_1(w_2))$. Thus, we can obtain $h(c, X, W) \neq h(c', X', W')$.

(2) If $c \neq c'$, we show $h(c, X, W) \neq h(c', X', W')$ by contradiction. Assume $h(c, X, W) = h(c', X', W')$, we have

$$
\begin{aligned}
(1+\epsilon)f_1(c) + \sum_{x \in X} f_1(x) + \sum_{\{\{w_1, w_2\}\} \in W} f_2(f_1(w_1) + f_1(w_2)) = \\
(1+\epsilon)f_1(c') + \sum_{x \in X'} f_1(x) + \sum_{\{\{w_1, w_2\}\} \in W'} f_2(f_1(w_1) + f_1(w_2)).
\end{aligned}
\tag{11}
$$

This can be rewritten as

$$
\begin{aligned}
\epsilon(f_1(c) - f_1(c')) = \Big( f_1(c') + \sum_{x \in X'} f_1(x) + \sum_{\{\{w_1, w_2\}\} \in W'} f_2(f_1(w_1) + f_1(w_2)) \Big) \\
- \Big( f_1(c) + \sum_{x \in X} f_1(x) + \sum_{\{\{w_1, w_2\}\} \in W} f_2(f_1(w_1) + f_1(w_2)) \Big).
\end{aligned}
\tag{12}
$$

Since $f_1(c) - f_1(c') \neq 0$ and it is rational, given $\epsilon$ is irrational, we can conclude that L.H.S. of Eq. (12) is irrational. R.H.S. of Eq. (12), however, is rational. This reaches a contradiction. Thus, if $c \neq c'$, we have $h(c, X, W) \neq h(c', X', W')$.

For any function $g$ on $(c, X, W)$, we can construct a function $\varphi$ such that $\varphi\big(h(c, X, W)\big) = g(c, X, W)$. This is always achievable since $h(c, X, W)$ is injective.

**Further justification for the first layer**. If the input features $x \in \mathcal{X}$ are one-hot encodings, $f_1$ is not necessary and thus can be removed. In other words, we can show as follows that there exists an $f_2$ such that $h'(c, X, W) = (1+\epsilon)c + \sum_{x \in X} x + \sum_{\{\{w_1, w_2\}\} \in W} f_2(w_1 + w_2)$ is unique for any distinct 3-tuple of $(c, X, W)$. Note that $\sum_{x \in X} x$ is injective if input features are one-hot encodings, and the value of $\sum_{x \in X} x$ must be composed of integers. In addition, $\psi'(\{\{w_1, w_2\}\}) = w_1 + w_2$ is also injective. Similarly, We define the set $\mathcal{Y}' = \{\psi'(\{\{w_1, w_2\}\}) | w_1, w_2 \in \mathcal{X}\}$. We consider $Z_2 : \mathcal{Y}' \to \mathbb{N}$ be an injection from $y \in \mathcal{Y}'$ to natural numbers and $f_2 = N^{-Z_2(y)}$, where $N > |W|$ for all $W$. Then $h'(c, X, W) = (1+\epsilon)c + \sum_{x \in X} x + \sum_{\{\{w_1, w_2\}\} \in W} f_2(w_1 + w_2)$ is injective, since $\sum_{\{\{w_1, w_2\}\} \in W} f_2(w_1 + w_2)$ is unique for any $W$ and is a number $\in (0, 1)$, thus differing from the integer-valued $\sum_{x \in X} x$. This is why we do not need another MLP to model $f_1^{(1)}$ in Eq. (6). $\qquad \square$

# B  EXPERIMENTAL DETAILS

## B.1  DATASET STATISTICS

Table 7: Dataset Statistics. Avg. #Message$_{NC}$ denotes the average #Message$_{NC}$ per node.

| Dataset | Task | Domain | #Graphs | Avg. #Nodes | Avg. #Edges | Avg. #Message$_{NC}$ |
|---------|------|--------|---------|-------------|-------------|----------------------|
| IMDB-B | Graph classification | Social network | 1000 | 19.7 | 96.5 | 59.5 |
| IMDB-M | Graph classification | Social network | 1500 | 13 | 131.8 | 70.6 |
| COLLAB | Graph classification | Social network | 5000 | 74.5 | 4915.6 | 5016.2 |
| PROTEINS | Graph classification | Bioinformatics network | 1113 | 39.1 | 145.6 | 2.1 |
| ogbg-ppa | Graph classification | Bioinformatics network | 78200/45100/34800 | 243.4 | 2266.1 | 179.3 |
| PATTERN | Node classification | Social network | 10000/2000/2000 | 118.9 | 6079.8 | 3440.1 |
| CLUSTER | Node classification | Social network | 10000/1000/1000 | 117.2 | 4303.8 | 1301.5 |

## B.2  MODEL CONFIGURATIONS AND TRAINING HYPERPARAMETERS

For efficiency, we do not tune the model configurations and training hyperparameters for NC-GNN extensively. Since our NC-GNN model is a natural extension of GIN, we usually use the model configurations and tuned hyperparameters of GIN from the comminity as the start point for NC-GNN and then tune them a little bit according to the validation results.

For the model architecture, we tune the following configurations; those are (1) the number of layers, (2) the number of hidden dimensions, (3) using jumping knowledge (JK) technique or not, and (4) using residual connection or not. To ensure fair comparison, we only consider employing techniques (3) and (4) on the datasets where the baseline GIN model also use them.

In terms of training, we consider tuning the following hyperparameters. those are (1) the initial learning rate, (2) the step size of learning rate decay, (3) the multiplicative factor of learning rate decay, (4) the batch size, (5) the dropout rate, and (6) the total number of epochs.

The selected model configurations and training hyperparameters for all datasets are summarized in Table 8. For each dataset from GNN Benchmark, we have several NC-GNN models under different parameter budgets, as described in Section 6. Accordingly, we list the configurations and hyperparameters for all of these models for reproducibility.

Table 8: The selected model configurations and training hyperparameters of NC-GNN on all datasets.

| Dataset | IMDB-B | IMDB-M | COLLAB | PROTEINS | ogbg-ppa | PATTERN | CLUSTER |
|---|---|---|---|---|---|---|---|
| # Layers | 5 | 5 | 5 | 5 | 5 | 4/4/16 | 4/4/16 |
| # Hidden Dim. | 64 | 64 | 64 | 32 | 300 | 70/154/78 | 70/154/78 |
| JK | ✓ | ✓ | ✓ | ✓ | - | ✓ | ✓ |
| Residual Con. | - | - | - | - | - | - | - |
| Initial LR | 0.001 | 0.005 | 0.005 | 0.001 | 0.01 | 0.001 | 0.001 |
| Step size of LR | 50 | 50 | 20 | 50 | 30 | 20 | 20 |
| Mul. fac. of LR | 0.5 | 0.5 | 0.5 | 0.5 | 0.1 | 0.5 | 0.5 |
| Batch size | 32 | 128 | 256 | 32 | 32 | 32 | 32 |
| Dropout rate | 0.5 | 0.5 | 0.5 | 0 | 0.5 | 0/0.1/0.1 | 0/0/0.5 |
| # Epochs | 200 | 300 | 100 | 100 | 80 | 100/140/140 | 100/100/200 |

### B.3 THE NC-GNN MODEL FOR GRAPHS WITH EDGE FEATURES

The layer-wise formulation of our NC-GNN model with considering edge features is

$$
\boldsymbol{h}_v^{(\ell)} = \text{MLP}_1^{(\ell)}\Big(\big(1+\epsilon^{(\ell)}\big)\boldsymbol{h}_v^{(\ell-1)} + \sum_{u\in\mathcal{N}_v}\text{RELU}(\boldsymbol{h}_u^{(\ell-1)}+\boldsymbol{e}_{uv}) + \boxed{\sum_{\substack{u_1,u_2\in\mathcal{N}_v\\(u_1,u_2)\in E}}\text{MLP}_2^{(\ell)}\big(\boldsymbol{h}_{u_1}^{(\ell-1)} + \boldsymbol{h}_{u_2}^{(\ell-1)} + \boldsymbol{e}_{u_1u_2}\big)}\Big),
$$

which is a natural extension of Eq. (6) by including edge features. $\boldsymbol{e}_{uv}$ is the edge feature associated with edge $(u,v)$. In practice, we usually apply an embedding layer to input edge features such that they have the same dimension as the node representations.

For reference, the GIN model with considering edge features (Hu et al., 2019) can be formulated as

$$
\boldsymbol{h}_v^{(\ell)} = \text{MLP}_1^{(\ell)}\Big(\big(1+\epsilon^{(\ell)}\big)\boldsymbol{h}_v^{(\ell-1)} + \sum_{u\in\mathcal{N}_v}\text{RELU}(\boldsymbol{h}_u^{(\ell-1)} + \boldsymbol{e}_{uv})\Big).
$$

