# OpenReview forum: "Your Neighbors Are Communicating: Towards Powerful and Scalable Graph Neural Networks"
_ICLR.cc/2023/Conference — Submitted to ICLR 2023_

### Official Review · Reviewer_xnLi · 2022-10-24

**Confidence:** 2
**Correctness:** 3
**Technical Novelty And Significance:** 3
**Empirical Novelty And Significance:** 3
**Recommendation:** 6

**Clarity, Quality, Novelty And Reproducibility:**

Clarity
- The problem setup and motivation are clear. The paper also provides details about its methodology.

Quality
- The writing quality of this paper overall is pretty decent.

Novelty
- The paper has good discussions in the introduction and related work about its novelty.

Reproducibility
- The paper does not provide limited details about its implementation and hyperparameter settings. The reviewer is not very confident of reproducing the results with the description.


**Strength And Weaknesses:**

Strengths:
- It is an interesting idea to explore how edges among neighbor nodes would affect the effectiveness of GNN.
- The proposed method is guided by theoretical analysis and the end results demonstrate promising accuracy improvements over existing methods.

Weaknesses:
- While getting promising accuracy results, the proposed method appears to have a higher memory complexity than existing ones. Therefore, it is less clear why the proposed method is more scalable than existing methods.
 - While getting higher accuracy under a similar amount of trainable parameters, the training time cost of NC-GNN is significantly higher than baseline methods such as GIN. For example, on Pattern and Cluster, NC-GNN incurs 4-6x longer training time, raising the question of whether the improved accuracy is at a cost of a much higher training budget. Also, notably, when the training budget is about the same (e.g., IMDB-B), the accuracy difference is minimal (with no statistical significance). This raises questions on whether the improvement can be achieved by simply training with a larger training budget and some hyperparameter tuning (e.g., learning rate schedules). Without this question answered, one cannot definitely say that considering neighbor node connections is the mere factor for the observed accuracy. Unfortunately, the paper makes no attempt to demonstrate whether that's the case. To be more convincing, it would be better for the authors to compare the proposed method with existing methods under the same training budget (not just the same amount of parameters but also a similar training time) with detailed hyperparameter setups.


**Summary Of The Paper:**

This paper studies the problem of improving the expressiveness of graph neural networks. In particular, the paper proposes NC-GNN, a GNN training method that incorporates edges among neighbors as additional information for improving the accuracy of GNN. The work also provides some theoretical justification on why the proposed method is more expressive than existing work such as the 1-dimensional WL algorithm, in terms of identifying non-isomorphic graphs. Evaluations of several data sets show that the proposed method can achieve higher accuracy than existing methods such as GCN, GraphSAGE, and GIN.

**Summary Of The Review:**

Despite there being some unanswered questions such as whether the improvement is at the cost of an increased training budget, the reviewer is overall positive about the work given the theoretical justification of its method and the promising accuracy results.


==============================
Post-rebuttal comments:

The reviewer has read the other reviewers' comments and the authors' responses. The reviewer appreciates the authors for adding additional evaluation results to show the benefits of reduced training time and memory usage than existing work such as GIN-AK+. From that result, the improvement indeed looks quite promising and large. On the other hand, the concerns raised by the reviewer Agxt and other reviewers, such as limited theoretical contribution and the evaluation methodology (e.g., metrics and missing datasets), also seem to be valid. That said, the reviewer finds this work exposes opportunities for further scaling GNN training, so I will keep my score positive.

---

> ### Author Response · Authors · 2022-11-14
> **Response from the authors**
>
> Thank you for the comments. We provide our response as below.
>
> > While getting promising accuracy results, the proposed method appears to have a higher memory complexity than existing ones. Therefore, it is less clear why the proposed method is more scalable than existing methods.
>
> - Our scalability means that our NC-GNN is more scalable compared to other methods aiming at improving the expressiveness of message passing GNNs. As analyzed in Table 1, our NC-GNN is indeed more scalable. Besides Table 5, we also added an additional comprehensive comparison to subgraph GNNs in Table 6 to empirically show the scalability of our method. According to the results, our NC-GNN is more efficient in terms of training and inference time, as well as memory usage.
>
> - Compared to original message passing GNNs, such as GIN, our NC-GNN has a bounded memory increasement and preserves the linear time complexity with a constant factor, as analyzed in Table 1. In summary, our method reaches a sweet spot between scalability and expressivity.
>
> > While getting higher accuracy under a similar amount of trainable parameters, the training time cost of NC-GNN is significantly higher than baseline methods such as GIN. For example, on Pattern and Cluster, NC-GNN incurs 4-6x longer training time, raising the question of whether the improved accuracy is at a cost of a much higher training budget. Also, notably, when the training budget is about the same (e.g., IMDB-B), the accuracy difference is minimal (with no statistical significance). This raises questions on whether the improvement can be achieved by simply training with a larger training budget and some hyperparameter tuning (e.g., learning rate schedules). Without this question answered, one cannot definitely say that considering neighbor node connections is the mere factor for the observed accuracy. Unfortunately, the paper makes no attempt to demonstrate whether that's the case. To be more convincing, it would be better for the authors to compare the proposed method with existing methods under the same training budget (not just the same amount of parameters but also a similar training time) with detailed hyperparameter setups.
>
> - As analyzed in our paper, **the improvement margin over GIN is related to the number of edges among neighbors in the graph** (statistically summarized in Table 7 as Avg. \#Message$_{NC}$). That is why our NC-GNN has significantly more obvious improvements over GIN on COLLAB, ogbg-ppa, PATTERN, and CLUSTER.
>
> - In most cases, the training time is not a main factor for performance improvement. To verify this, as suggested, we conduct a simple experiment to compare NC-GNN with GIN under the same training time budget on PATTERN. Specifically, we increase the hidden dimension of GIN and training GIN under the same time budget as NC-GNN. As shown below, the obtained improvement of NC-GNN over GIN demonstrates that the improvement cannot be achieved by simply training with a larger training time budget. This verifies that the improvements obtained by NC-GNN are actually come from modeling edges among neighbors, which is consistent with our theoretical results.
>
> |GIN|NC-GNN|
> |----|----|
> |84.73+/-0.007|**86.73**+/-0.007|
>
> > Reproducibility
> - Our code will be published with our paper.
> -----
> Hope you addressed the questions well. Looking forward to any further feedback. Thank you!

---

### Official Review · Reviewer_zieR · 2022-10-25

**Confidence:** 3
**Correctness:** 4
**Technical Novelty And Significance:** 3
**Empirical Novelty And Significance:** 3
**Recommendation:** 5

**Clarity, Quality, Novelty And Reproducibility:**

Overall the quality of the article is good, although the comparison in the experiments is not sufficient. Clarity is fine. I cannot evaluate the originality of the work because I am only knowledgeable on the k-WL problem.

**Strength And Weaknesses:**

Strength:
1. This method is simple and efficient.  The superiority of our NC-1-WL lies in improving the expressiveness over 1-WL, while being
efficient as 1-WL.

2. NC-GNN is provably as powerful as NC-1-WL in distinguishing non-isomorphic graphs.

Weakness:
Although this paper mainly focuses on a GNN for the large-scale dataset. I think a comprehensive comparison is necessary even if the NC-GNN‘s performance is not as good as others. GraphSNN is only mentioned in this paper but do not compare in the experiment.  Comparison of real training time also needs to add other methods. In general, I felt that the experimentation was rather inadequate and seemed to deliberately avoid certain methods. This, therefore, reduces the convincing of the experiment.

In fact and I'm curious as to why NC-GNN didn't try it on the node classification task. In my opinion,

**Summary Of The Paper:**

In this paper, based on message-passing frameworks, the authors propose a new graph neural network with expressiveness upper-bounded by 3-WL theoretically. Instead of aggregating the 1-hop neighbours, NC-GNN considers the edge between 1-hop neighbours. More importantly, the implementation is very efficient and thus is able to run in large-scale datasets. Experiments demonstrate that the proposed NC-GNN achieves good performance on various benchmarks.

**Summary Of The Review:**

As I'm not an expert in the field, I can't give an opinion at this stage as I don't see any hard points. I therefore prefer to refer to other reviewers' comments.

---

> ### Author Response · Authors · 2022-11-14
> **Additional empirical results are added**
>
> Thank you for the review. We provide our response as below.
>
> > Although this paper mainly focuses on a GNN for the large-scale dataset. I think a comprehensive comparison is necessary even if the NC-GNN‘s performance is not as good as others. GraphSNN is only mentioned in this paper but do not compare in the experiment. Comparison of real training time also needs to add other methods. In general, I felt that the experimentation was rather inadequate and seemed to deliberately avoid certain methods. This, therefore, reduces the convincing of the experiment.
> - As suggested, we have added the results of GraphSNN on TUDatasets to Table 1. GraphSNN performs better than NC-GNN on two small datasets PROTEINS and IMDB-B. On the large-scale graph classification over ogbg-ppa, our NC-GNN outperforms GraphSNN as shown in the following table.
> |Method|`GraphSNN`|`NC-GNN`|
> |----|----|----|
> |Test Acc|70.66+/-1.65|**71.94**+/-0.43|
> - We also add an experiment to comprehensively compare with subgraph GNNs. Specifically, we compare to GNN-AK+, a representative method in the subgraph GNN family, on **test accuracy**, **training time per epoch**, **total training time for convergence**, **GPU memory usage**, **MACS**, and **inference time**.  MACS is a metric calculating the average multiply-accumulate operations for each graph in the test set. The detailed setup is included in our revision. The results are shown below and included in our revision. On both PATTERN and CLUSTER, our NC-GNN achieves competitive accuracies as GNN-AK+, while being more efficient in training, including training time per epoch and total training time. In addition, we use less GPU memory since we do not have to consider updating node representations for all the nodes in the expanded subgraphs as GNN-AK+. More importantly, the MACS overhead of GNN-AK+ is 100x more than our NC-GNN. Since our NC-GNN calculates each node representation from the original graph instead of the expanded subgraphs, it can save huge MACS overhead during the forward procedure. To further show the advantage of fewer MACS overhead, we provide the inference time comparison and our NC-GNN takes less time during inference. Overall, NC-GNN reaches a sweet spot between expressivity and scalability, from both theoretical and practical observations.
>
> |Dataset|Method|# Layers|# Param|`Test Acc.`|`Time/epoch`|`Total Time`|`GPU Memory`|`Macs`|`Inference Time`|
> |----|----|----|----|----|----|----|----|----|----|
> |PATTERN|`GIN-AK+`|4|601136|**86.836** +/-0.007|77.503s|0.667h|31434 MB|176.445G|32.188s|
> |PATTERN|`NC-GNN`|4|552096|86.71+/-0.069|**42.517s**|**0.555h**|**15625 MB**|**1.142G**|**12.025s**|
> |CLUSTER|`GIN-AK+`|16|602586|76.502+/-0.210|148.983s|1.420h|32142 MB|110.518G|27.163s |
> |CLUSTER|`NC-GNN`|16|562948|**76.992**+/- 0.063|**48.869s**|**0.679h**|**22386 MB**|**0.841G**|**5.1763s**|
>
>
> > In fact and I'm curious as to why NC-GNN didn't try it on the node classification task. In my opinion
>
> - We have conducted experiments on node classification tasks to evaluate our NC-GNN. Note that the PATTERN and CLUSTER datasets in Table 4 correspond to node classification tasks. Our NC-GNN shows significant improvements over GIN. This can directly demonstrate the effectiveness of modeling the information of edges among neighbors, since the only difference of NC-GNN to GIN is the third term in Eq. (6). This strongly shows our NC-GNN is simple and effective.
>
> ----
> Hope you addressed the questions well. Please let us know if there are any further feedback. Thank you!

---

> ### Author Response · Authors · 2022-11-29
> **Looking forward to your response**
>
> Dear Reviewer zieR, could you take a look at our response and let us know if the questions are well addressed? Any further questions and comments are warmly welcome. Thank you!

---

### Official Review · Reviewer_nYWZ · 2022-10-28

**Confidence:** 3
**Clarity, Quality, Novelty And Reproducibility:** 1. The paper is well-written, and I b…
**Correctness:** 2
**Technical Novelty And Significance:** 3
**Empirical Novelty And Significance:** 2
**Recommendation:** 5

**Strength And Weaknesses:**

#### Strengths
1. To the best knowledge, I think the construction of this modified graph isomorphism test, NC-1-WL is novel. And I like the idea of further modeling the "edge coloring" as multisets of multisets, which seems naturally extend the WL-test algorithm. It is not hard to understand its superiority to the standard 1-WL test, while it still only relies on only local operations, in contrast to 3-WL.
2. The experimental results compared with GIN demonstrate that this added embedding from pairs of neighbors is effective for real-world datasets and not only graph classification but also node classification tasks.
3. The paper writing is very polished and clear.
#### Weaknesses
1. I think the experimental comparison to subgraph-based GNNs is very insufficient in the current manuscript. The authors only show the performance comparison to subgraph-based GNNs on the TUDatasets, which are mostly small graphs and are gradually replaced by OGB (Hu et al., 2020) and GNN Benchmark (Dwivedi et al., 2020). Thus it would be interesting and important to see more comparisons to subgraph-based baselines on those datasets. More importantly, one major advantage of the proposed NC-GNN is the scalability/efficiency. So the authors should expand Table 5. to compare with subgraph-based GNNs in terms of the epoch-time, memory usage, convergence speed, etc.
2. I think the related work section needs some expansion to discuss more prior papers (apart from subgraph-based GNNs) which (1) propose more powerful GNNs, maybe not in the sense that strictly better than 1-WL, while maintaining the scalability (e.g., [1]) or (2) follow the WL-hierarchy and are probably better than 1-WL, while being much more efficient than mimicking the k-WL test algorithm (e.g. [2]). Recently there have been many papers trying to tackle these two directions, and I hope the authors could do a more careful literature review of those papers.

#### Minor Issues
1. I suggest the authors color the reference links.

##### References
- [1] Papp, P. A., Martinkus, K., Faber, L., & Wattenhofer, R. (2021). Dropgnn: random dropouts increase the expressiveness of graph neural networks. Advances in Neural Information Processing Systems, 34, 21997-22009.
- [2] Puny, O., Ben-Hamu, H., & Lipman, Y. (2020). Global attention improves graph networks generalization. arXiv preprint arXiv:2006.07846.

**Summary Of The Paper:**

This paper tackles the challenging goal: developing a general GNN framework with provably expressive power while maintaining the scalability of the message-passing scheme. The authors first model the edges among neighbors as a multiset of multisets and formulate the NC-1-WL graph isomorphism. They show that the expressiveness of NC-1-WL in distinguishing non-isomorphic graphs is strictly above 1-WL and below 3-WL. But based on NC-1-WL, they can propose a differentiable version called NC-GNN, which is intrinsically more scalable than the algorithms that mimic the WL graph isomorphism test. Experimental results on not only graph classification but also node classification tasks show that NC-GNN can consistently outperform GIN by significant margins on various tasks and achieve competitive performance as existing expressive GNNs while being much more efficient.

**Summary Of The Review:**

Overall for the current manuscript, I recommend weak rejection. I like the idea of the proposed NC-1-WL test and the corresponding NC-GNN algorithm. However, since recently there have been many papers on provably powerful while scalable GNNs, the current insufficient experimental comparison to those work, including subgraph-based GNNs, on larger graphs, and in terms of efficiency, cannot support the claimed contributions of NC-GNN well.

---

> ### Author Response · Authors · 2022-11-14
> **Added the suggested experimental comparison**
>
> Thank you for the feedback.
>
> > I think the experimental comparison to subgraph-based GNNs is very insufficient in the current manuscript. The authors only show the performance comparison to subgraph-based GNNs on the TUDatasets, which are mostly small graphs and are gradually replaced by OGB (Hu et al., 2020) and GNN Benchmark (Dwivedi et al., 2020). Thus it would be interesting and important to see more comparisons to subgraph-based baselines on those datasets. More importantly, one major advantage of the proposed NC-GNN is the scalability/efficiency. So the authors should expand Table 5. to compare with subgraph-based GNNs in terms of the epoch-time, memory usage, convergence speed, etc.
> - As suggested, to comprehensively compare with subgraph GNNs, we added an additional experiment on the used two large-scale node classification tasks. Specifically, we compare to GNN-AK+, a representative method in the subgraph GNN family, on **test accuracy**, **training time per epoch**, **total training time for convergence**, **GPU memory usage**, **MACS**, and **inference time**.  MACS is a metric calculating the average multiply-accumulate operations for each graph in the test set. The detailed setup is included in our revision. The results are shown below and included in our revision. On both PATTERN and CLUSTER, our NC-GNN achieves competitive accuracies as GNN-AK+, while being more efficient in training, including training time per epoch and total training time. In addition, we use less GPU memory since we do not have to consider updating node representations for all the nodes in the expanded subgraphs as GNN-AK+. More importantly, the MACS overhead of GNN-AK+ is 100x more than our NC-GNN. Since our NC-GNN calculates each node representation from the original graph instead of the expanded subgraphs, it can save huge MACS overhead during the forward procedure. To further show the advantage of fewer MACS overhead, we provide the inference time comparison and our NC-GNN takes less time during inference. Overall, NC-GNN reaches a sweet spot between expressivity and scalability, from both theoretical and practical observations.
>
> |Dataset|Method|# Layers|# Param|`Test Acc.`|`Time/epoch`|`Total Time`|`GPU Memory`|`Macs`|`Inference Time`|
> |----|----|----|----|----|----|----|----|----|----|
> |PATTERN|`GIN-AK+`|4|601136|**86.836** +/-0.007|77.503s|0.667h|31434 MB|176.445G|32.188s|
> |PATTERN|`NC-GNN`|4|552096|86.71+/-0.069|**42.517s**|**0.555h**|**15625 MB**|**1.142G**|**12.025s**|
> |CLUSTER|`GIN-AK+`|16|602586|76.502+/-0.210|148.983s|1.420h|32142 MB|110.518G|27.163s |
> |CLUSTER|`NC-GNN`|16|562948|**76.992**+/- 0.063|**48.869s**|**0.679h**|**22386 MB**|**0.841G**|**5.1763s**|
>
>
> > I think the related work section needs some expansion to discuss more prior papers (apart from subgraph-based GNNs) which (1) propose more powerful GNNs, maybe not in the sense that strictly better than 1-WL, while maintaining the scalability (e.g., [1]) or (2) follow the WL-hierarchy and are probably better than 1-WL, while being much more efficient than mimicking the k-WL test algorithm (e.g. [2]). Recently there have been many papers trying to tackle these two directions, and I hope the authors could do a more careful literature review of those papers.
> - We have added the review on the suggested related work in our revision. Our current Sec 5 does a thorough review of related work.
>
> > I suggest the authors color the reference links.
> - We have colored the reference links in our revision.
>
> > Reproducibility
> - Our code will be published with our paper.
> --------
> Thank you for the raised questions and suggestions. Hope we addressed the questions well. Please let us know if there are any further questions.

---

> ### Author Response · Authors · 2022-11-29
> **Looking forward to your response**
>
> Dear Reviewer nYWZ, could you take a look at our response and let us know if the questions are well addressed? Any further questions and comments are warmly welcome. Thank you!

---

### Official Review · Reviewer_Agxt · 2022-10-28

**Confidence:** 5
**Clarity, Quality, Novelty And Reproducibility:** Presentation is clear.
**Correctness:** 3
**Technical Novelty And Significance:** 1
**Empirical Novelty And Significance:** 1
**Recommendation:** 3

**Strength And Weaknesses:**

**Strength**:
1. Simple, and fast.
2. Effective in several real-world datasets.

**Weakness**:
1. Not novel. The design is too incremental comparing with these Subgraph GNNs. Although being simple and fast.
2. It is just a special case of recently neurips 2022 paper [How Powerful are K-hop Message Passing Graph
Neural Networks], with K=1. The NeurIPS paper is general and with extensive theoretical analysis. In this paper I can only find the simplicity as a contribution. The incremental design doesn't help the research area.
3. All theorems are mentioned in existing papers. The 3-WL bound is inside [Understanding and extending subgraph gnns by rethinking their symmetries].
4. Experimental evaluation misses too many baselines. First, we all know that TUDataset is not good for evaluating expressivity. Second, widely used ZINC is missing. Third, performance of NestedGNN, GNN-AK, ESAN are not inside the paper. Of course the performance shouldn't be higher than them, but the author should include and also include run-time comparison.


**Summary Of The Paper:**

This paper proposes a simple extension to current message passing based GNN, inspired from recent subgraph GNNs. Although being less powerful than subgraph GNNs, the proposed method is efficient with limited computation overhead comparing with message passing GNNs. The author demonstrates its improvements over many real-world datasets.

**Summary Of The Review:**

The author proposes a simple add-on to the current message passing based GNN with the same intuition mentioned in [Zhao et al. 2021] but more much efficient implementation (with being less powerful), however this efficient implementation is also covered by a recent paper [Feng et al. 2022] . Given all these existing works, I can hardly find enough contribution in this paper. (If this paper is designed 1-year eariler it's possible to be novel, however currently I really cannot find novelty.)




[Zhao et al. 2021] From stars to subgraphs: Uplifting any GNN with local structure awareness.
[Feng et al. 2022] How Powerful are K-hop Message Passing Graph Neural Networks

---

> ### Author Response · Authors · 2022-11-14
> **Response (Part 2)**
>
> > Experimental evaluation misses too many baselines. First, we all know that TUDataset is not good for evaluating expressivity. Second, widely used ZINC is missing. Third, performance of NestedGNN, GNN-AK, ESAN are not inside the paper. Of course the performance shouldn't be higher than them, but the author should include and also include run-time comparison.
> - Although TUDatasets are small datasets, it is widely used as a benchmark by previous expressive GNNs, including the subgraph GNNs (such as GNN-AK) mentioned by the reviewer. We **include the experiments on TUDatasets for completeness**. Having said this, we still observed obvious improvements that are consistent to our main claim. Specifically, NC-GNN achieves a notable improvement margin of 2.3 on COLLAB over GIN. This aligns with our claim since COLLAB has a considerable number of edges among neighbors, as presented in Table 7.
>
> - We also noticed that TUDatasets are small. That is why we also perform the experiments on large-scall graph classification dataset (Table 3) and node classification datasets (Table 4). The results can strongly support our main claim of including edges among neighbors.
>
> - As described in our paper (below Eq. (6)), if there does not exist any edges among neighbors for all nodes in a graph, our model will reduce to the GIN model. We found that the graphs in ZINC do not have such edges. In that case, our method is not applicable.
> - To comprehensively compare with subgraph GNNs, we added an additional experiment on the used two large-scale node classification tasks. Specifically, we compare to GNN-AK+, a representative method in the subgraph GNN family, on **test accuracy**, **training time per epoch**, **total training time for convergence**, **GPU memory usage**, **MACS**, and **inference time**.  MACS is a metric calculating the average multiply-accumulate operations for each graph in the test set. The detailed setup is included in our revision. The results are shown below and included in our revision. On both PATTERN and CLUSTER, our NC-GNN achieves competitive accuracies as GNN-AK+, while being more efficient in training, including training time per epoch and total training time. In addition, we use less GPU memory since we do not have to consider updating node representations for all the nodes in the expanded subgraphs as GNN-AK+. More importantly, the MACS overhead of GNN-AK+ is 100x more than our NC-GNN. Since our NC-GNN calculates each node representation from the original graph instead of the expanded subgraphs, it can save huge MACS overhead during the forward procedure. To further show the advantage of fewer MACS overhead, we provide the inference time comparison and our NC-GNN takes less time during inference.
>
> |Dataset|Method|# Layers|# Param|`Test Acc.`|`Time/epoch`|`Total Time`|`GPU Memory`|`Macs`|`Inference Time`|
> |----|----|----|----|----|----|----|----|----|----|
> |PATTERN|`GIN-AK+`|4|601136|**86.836** +/-0.007|77.503s|0.667h|31434 MB|176.445G|32.188s|
> |PATTERN|`NC-GNN`|4|552096|86.71+/-0.069|**42.517s**|**0.555h**|**15625 MB**|**1.142G**|**12.025s**|
> |CLUSTER|`GIN-AK+`|16|602586|76.502+/-0.210|148.983s|1.420h|32142 MB|110.518G|27.163s |
> |CLUSTER|`NC-GNN`|16|562948|**76.992**+/- 0.063|**48.869s**|**0.679h**|**22386 MB**|**0.841G**|**5.1763s**|
>
>
> ---------------
> Overall, our contributions are the **powerful** and **scalable** NC-1-WL and NC-GNN. We achieved a **sweet spot between expressivity and scalability, from both theoretical and practical observations.**. Because of the simplicity and expressivity, we anticipate that NC-GNN will become an important base model for learning from graphs, especially social network graphs. We hope the reviewer can value this achievement. Please let us know if there are any further concerns. Thank you.
>
> [1] Zhang, Muhan, and Pan Li. "Nested graph neural networks." Advances in Neural Information Processing Systems 34 (2021): 15734-15747.
>
> [2] Zhao, Lingxiao, et al. "From Stars to Subgraphs: Uplifting Any GNN with Local Structure Awareness." International Conference on Learning Representations. 2021.

---

> > ### Comment · Reviewer_Agxt · 2022-11-14
> > **Response to author (2nd Round)**
> >
> > Thank you for providing clarification to several questions and adding additional result.
> > From the comparison time on PATTERN and CIFAR, the proposed method is yet simple and fast.
> > However, I still have concerns over novelty and the experimental section.
> >
> > > Our method is related to but fundamentally different from subgraph GNNs (such as [1][2]).
> >
> > If I'm correct, the GNN-AK paper introduces subgraph-1-WL algorithm in terms of isomorphism test,
> > which introduces the idea of encoding rooted subgraph with a graph hashing function. The NC-1-WL
> > is a realization of subgraph-1-WL with using 1-hop rooted subgraph, and a simple 1-hop rooted subgraph
> > encoding function: encodes all colors of nodes and edges in the 1-hop rooted subgraph. Hence in the root,
> > the fundamental intuition and idea is the same. Given that GNN-AK is published 1 year ago, the fundamental idea
> > is not novel, but however the simplicity is the contribution of this paper.
> >
> >
> >
> > > After a careful reading, we agree that our work is related to this NeurIPS2022 paper. However, there are still key differences.
> >
> > * The paper is available online in arxiv on 5/26/2022. Based on ICLR review guideline, only papers after 5/28/2022 are considered contemporary work. In this case, the paper is not contemporary work and should be discussed thoroughly.
> > * By checking the NeurIPS paper, the paper extends the k-hop gnn based aggregation with additional "peripheral subgraph".
> > And when k=1, the peripheral subgraph edge set $Q_{v,G}^{1,t}$ is the edge set considered inside the author's design. The equation (5)
> > of the NeurIPS paper is the general formulation of how to encoding the peripheral subgraph. To improve simplicity, the NeurIPS paper proposes to use the number of edges as the encoding of the edge set.  In fact: 1) again, this paper's realization is a special case of equation (5). 2) It seems that encoding with the number of edges of the 1-hop subgraph should be even faster than the author's realization, which questions whether the main contribution, simplicity, is really the most simple design.  Given that, I think the author should **compare with encoding with number of edges directly**, as this is a even simpler baseline than the author's design. If there is not much improvement comparing with encoding with number of edges, then even the simplicity contribution will be invalid.
> > * The author first argues that the isomorphism test NC-1-WL is another contribution on top of NC-1-GNN. I can hardly agree, as the field always proposes WL and corresponding GNN together as a single contribution (thanks to GIN's theoretical result of connecting GNN and WL).
> >
> > > Novelty of theorem.
> >
> > In fact, as the paper is a special case of equation (5) of [How Powerful are K-hop Message Passing Graph
> > Neural Networks], which also includes the theorem of lower bound of 1-WL and upper bound of 3-WL, all theorems are directly implied by the NeurIPS paper.
> >
> >
> > **Further experimental results are needed.**
> >
> > Given that simplicity and fast is the main contribution, the author really need to demonstrate the effectiveness of the proposed paper (comparable performance and significant runtime reduction) on extensive datasets. The current included datasets are subsets of GNN-AK and KP-GNN (NeurIPS paper). The author at least have to show:
> >
> > **1)** Comparable number of datasets with existing literature (actually I think even more datasets are needed to show strength the main contribution: the simplicity for practical usage).
> >  **2)** Comparable performance with respect to GNN-AK and KP-GNN.
> > **3)** Great time reduction comparing with GNN-AK and KP-GNN.
> > **4)** Performance improvement over using encoding with number of edges inside 1-hop rooted subgraph.

---

> > > ### Author Response · Authors · 2022-11-14
> > > **Authors' response (part 2)**
> > >
> > > >  All theorems are directly implied by the NeurIPS paper.
> > >
> > > - Note that the proof of 1-WL lower bound and 3-WL upper bound in the NeurIPS paper is for K-hop message passing, not for KP-GNN. Our 1-WL lower bound and 3-WL upper bound theorems are strictly for NC-GNN, and the proofs are accordingly different, as detailed in Appendix A.2.
> > >
> > > - As clarified above, the existence of the NeurIPS paper should not be a valid reason to reduce our contribution, according to the ICLR reviewer guideline. We have discussed this contemporary work in our revision. In summary, our different contributions include (1) a new graph isomorphism test algorithm, NC-1-WL and (2) a simple, scalable, and powerful (both theoretically and empirically) NC-GNN model.
> > >
> > > > The author first argues that the isomorphism test NC-1-WL is another contribution on top of NC-1-GNN. I can hardly agree, as the field always proposes WL and corresponding GNN together as a single contribution (thanks to GIN's theoretical result of connecting GNN and WL).
> > > - We do not agree that it is always the case that new isomorphism test algorithms can be proposed with expressive GNNs. In contrast, most expressive GNNs do not correspond to new isomorphism test algorithms, such as NestGNN [1], GraphSNN [2] and ESAN[3]. It is non-trivial to have a rigorous and new isomorphism test algorithm. Therefore, we argue that **a new isomorphism test algorithm is an important contribution** (This contribution is also recognized by Reviewer nYWZ and Reviewer zieR). The previous works with proposed new isomorphism test algorithms also claim the contribution of the new isomorphism test algorithm [e.g., 4,5,6].
> > >
> > >
> > >
> > > > Further experimental results are needed.
> > > - Performing extensive comparison to KP-GNN should not be a requirement for our submission, according to the ICLR reviewer guide.
> > > - We would like to recap our main claim on our NC-GNN model: **our NC-GNN model is a scalable and powerful model.** Our goal is not to outperform most existing expressive (but expensive) GNNs. Overall, our experiments need to support the following two claims. (1) **Powerful**: Our NC-GNN can be practically more effective than regular message passing GNNs, such GIN. (2) **Scalable**: Our overhead over GIN should be reasonable and NC-GNN can be more efficient compared to existing expressive GNNs, while achieving competitive practical performance.
> > > - Our results in Table 2, 3, & 4 can support (1) strongly, while the results in Table 2, 4, 5, & 6 can verify (2) directly.
> > >
> > > [1] Zhang, Muhan, and Pan Li. "Nested graph neural networks." Advances in Neural Information Processing Systems 34 (2021): 15734-15747.
> > >
> > > [2] Wijesinghe, Asiri, and Qing Wang. "A New Perspective on" How Graph Neural Networks Go Beyond Weisfeiler-Lehman?"." International Conference on Learning Representations. 2021.
> > >
> > > [3] Bevilacqua, Beatrice, et al. "Equivariant Subgraph Aggregation Networks." International Conference on Learning Representations. 2021.
> > >
> > > [4] Morris, Christopher, Gaurav Rattan, and Petra Mutzel. "Weisfeiler and Leman go sparse: Towards scalable higher-order graph embeddings." Advances in Neural Information Processing Systems 33 (2020): 21824-21840.
> > >
> > > [5] Morris, Christopher, et al. "SpeqNets: Sparsity-aware Permutation-equivariant Graph Networks." arXiv preprint arXiv:2203.13913 (2022).
> > >
> > > [6] Zhao, Lingxiao, et al. "A Practical, Progressively-Expressive GNN." arXiv preprint arXiv:2210.09521 (2022).

---

> > > > ### Comment · Reviewer_Agxt · 2022-11-14
> > > > **Final Response to Author**
> > > >
> > > > I appreciate the author's additional response. Based on your response, we have agreement on two things:
> > > >
> > > > 1. NC-1-WL is a special case of Subgraph-1-WL of [1]. (Share same concept)
> > > > 2. NC-GNN is a special case of Equation (5) of [2]
> > > >
> > > > We have main conflict on **whether enough datasets are used for evaluation**.
> > > > Again, the used datasets are too limited comparing with [1] and [2]. Actually, many important datasets of reflecting expressivity are not used at all. Like counting substructures, predict graph properties, ZINC (the author claim no improvement on ZINC, as they said there is no triangle substructure inside, which I'm not sure), CIFAR10 (a larger graph classification dataset), and OGBG node classification datasets (the proposed algorithm should be useful for large scale node classification, as they claim the simplexity and fast). Also the TUDatasets used are just a subset of widely used tudatasets, which I'm not sure is picked based on performance or not.
> > > >
> > > > **Based on my evaluation, the novelty is limited, and experiments are not enough. I leave the decision (like whether considering [2] as contemporary not) to area chair for considering these factors.**
> > > >
> > > > [1] Zhao, Lingxiao, et al. "From stars to subgraphs: Uplifting any GNN with local structure awareness." arXiv preprint arXiv:2110.03753 (2021).
> > > > [2] Feng, Jiarui, et al. "How Powerful are K-hop Message Passing Graph Neural Networks." arXiv preprint arXiv:2205.13328 (2022).

---

> > > > > ### Author Response · Authors · 2022-11-14
> > > > > **Intuition clarification; experiments are designed to support the claims without cherry-picking**
> > > > >
> > > > > Thank you for the response.
> > > > > > Two agreements
> > > > >
> > > > > We respectfully disagree with the two agreements summarized by the reviewer.
> > > > > - We do share the similar intuition of considering the neighborhood subgraph. Actually, this intuition of considering the neighborhood subgraph has been proposed before the GNN-AK paper, such as in Ego-GNN [1], and widely considered by many existing works, such as [2, 3, 4]. Considering the similar intuition should not be treated as “a special case”. **We do not claim that this intuition is our contribution. Our contribution is to simply model the edges among neighbor as multisets of multisets, leading to the efficient and powerful NC-1-WL graph isomorphism test algorithm and the NC-GNN neural model.** The efficiency and expressive power has been supported both theoretically and empirically in our paper.
> > > > >
> > > > > - We do have overlap with the contemporary NeurIPS work [5]. We leave the contemporary work consideration to Area Chair, and would like to clarify **the additional contributions** as follows. **(1) We have additionally delivered a graph isomorphism test algorithm NC-1-WL, which is more expressive than 1-WL. (2) We have our dedicatedly designed NC-GNN model with rigorously proved expressiveness.**
> > > > >
> > > > > > Experiments
> > > > > - Since this research direction is popular, **we can hardly compare to all existing expressive GNNs on all datasets covered by previous works**. Our experiments are designed to support our main claims in our use cases (as described in the next point); these are (1)**powerful** (i.e., how the empirical performance compares to GIN and existing expressive GNNs) and (2)**scalable** (i.e., how the efficiency compares to GIN and existing expressive GNNs). We believe our comprehensive results in Table 2-6 can support these two claims well.
> > > > >
> > > > > - We **do not pick datasets based on performance**. As clearly pointed out in our paper (below Eq. (6) and at the beginning of Sec 6), **if there does not exist any edges among neighbors for all nodes in a graph, our model will reduce to the GIN model. Hence, we omit the dataset with the metric *Avg. #Message$_{NC}$* < 0.2, since the performance is almost the same as GIN**. We provide the metric statistics of some datasets mentioned by the reviewer in the following table. They do have rare edges among neighbors. The Avg. #Message_{NC} for the datasets we used are summarized in Table 7.
> > > > >
> > > > > |Dataset|REDDITBINARY| REDDITMULTI|MUTAG|PTC|NCI1|ZINC|
> > > > > |----|----|----|----|----|----|----|
> > > > > |Avg. #Message$_{NC}$|0.17|0.12|0|0.005|0.0045|0.0008|
> > > > >
> > > > > -----
> > > > > In summary, **we do have clearly clarified the use case of our method** and **demonstrate the effectiveness of our method in the desirable use cases**. We don’t think every dataset used by previous work should be treated as a golden testbed for our work.
> > > > >
> > > > > [1] Dylan Sandfelder, Priyesh Vijayan, and William L Hamilton. Ego-gnns: Exploiting ego structures in graph neural networks. In ICASSP 2021-2021 IEEE International Conference on Acoustics, Speech and Signal Processing (ICASSP), pp. 8523–8527. IEEE, 2021.
> > > > >
> > > > > [2] Zhang, Muhan, and Pan Li. "Nested graph neural networks." Advances in Neural Information Processing Systems 34 (2021): 15734-15747.
> > > > >
> > > > > [3] Zhao, Lingxiao, et al. "From stars to subgraphs: Uplifting any GNN with local structure awareness." arXiv preprint arXiv:2110.03753 (2021).
> > > > >
> > > > > [4] Wijesinghe, Asiri, and Qing Wang. "A New Perspective on" How Graph Neural Networks Go Beyond Weisfeiler-Lehman?"." International Conference on Learning Representations. 2021.
> > > > >
> > > > > [5] Feng, Jiarui, et al. "How Powerful are K-hop Message Passing Graph Neural Networks." arXiv preprint arXiv:2205.13328 (2022).

---

> > > ### Author Response · Authors · 2022-11-14
> > > **Further clarification on contribution, ICLR policy, and experiments. (Part 1)**
> > >
> > > Thank you for the response.
> > > > If I'm correct, the GNN-AK paper introduces subgraph-1-WL algorithm in terms of isomorphism test, which introduces the idea of encoding rooted subgraph with a graph hashing function. The NC-1-WL is a realization of subgraph-1-WL with using 1-hop rooted subgraph, and a simple 1-hop rooted subgraph encoding function: encodes all colors of nodes and edges in the 1-hop rooted subgraph. Hence in the root, the fundamental intuition and idea is the same. Given that GNN-AK is published 1 year ago, the fundamental idea is not novel, but however the simplicity is the contribution of this paper.
> > >
> > > We have clarified the difference with Subgraph-1-WL in the last paragraph in Sec 3. Specifically, Subgraph-1-WL **ideally** generalizes 1-WL from mapping the neighborhood to mapping the rooted subgraph. However, **injectively mapping the rooted subgraph (even the 1-hop rooted subgraph) is as difficult as graph isomorphism test problem and cannot be achieved**. In contrast, we simply consider modeling the edges among neighbors as a multiset of multisets, thus **achieving injectively mapping**. This difference makes our **NC-1-WL to be an executable algorithm for graph isomorphism testing**, compared to the Subgraph-1-WL concept. This simple design also helps us keep the locality of message passing, thus avoiding updating the representation of nodes from all expanded subgraphs as GNN-AK and leading to **the great balance between simplicity/scalability and expressiveness**.
> > >
> > > > The paper is available online in arxiv on 5/26/2022. Based on ICLR review guideline, only papers after 5/28/2022 are considered contemporary work. In this case, the paper is not contemporary work and should be discussed thoroughly.
> > >
> > > According to the ICLR reviewer guide (https://iclr.cc/Conferences/2023/ReviewerGuide), the NeurIPS2022 paper **is actually a contemporary work**.
> > > We quote ICLR review guideline as below:
> > >
> > > "Q: Are authors expected to cite and **compare with very recent work?** What about non peer-reviewed (e.g., ArXiv) papers? (updated on 7 November 2022)"
> > >
> > > "A: We consider papers contemporaneous if they are published (available in online proceedings) within the last four months. That means, since our full paper deadline is September 28, if a paper was published (i.e., **at a peer-reviewed venue**) on or after May 28, 2022, authors are not required to compare their own work to that paper. **Authors are encouraged to cite and discuss all relevant papers, but they may be excused for not knowing about papers not published in peer-reviewed conference proceedings or journals, which includes papers exclusively available on arXiv.** Reviewers are encouraged to use their own good judgement and, if in doubt, discuss with their area chair."
> > >
> > > Note that the mentioned paper was posted on arXiv on 5/26/2022 and accepted to NeurIPS 2022, whose proceeding is not published yet now. According to ICLR review guidelines, we are excused for not knowing this work and we are not supposed to compare with this work. Thus the novelty of our work should be evaluated without considering this paper. We understand that not every reviewer is fully aware of ICLR review guidelines and hope this reviewer can re-evaluate our work based on the above guideline.

---

> ### Author Response · Authors · 2022-11-14
> **Clarified our contributions; Added thorough comparison with subgraph GNNs; Our method reached a sweet spot between expressivity and scalability. (Part 1)**
>
> Thank you for the comments. We provide our response as below.
>
> > Not novel. The design is too incremental comparing with these Subgraph GNNs. Although being simple and fast.
> - Our method is **related to but fundamentally different from** subgraph GNNs (such as [1][2]). Subgraph GNNs [1][2] apply a base GNN to encode the neighborhood subgraph information of each node and then employs another GNN on the subgraph-encoded representations. Hence, subgraph GNNs are nested architectures. In contrast, we keep the locality/scalability of message passing. In other words, we only update the node representations for $n$ nodes as message passing GNNs instead of all nodes in all the expanded subgraphs. This fundamental difference brings the efficiency of our method, compared to subgraph GNNs, as shown by the complexity analysis in Table 1. We also compared to subgraph GNNs comprehensively in our following added experiments, which empirically demonstrate the competitive empirical performance and better efficiency compared to subgraph GNNs.
>
> - We respectfully argue that **being simple is non-trivial and is a significant contribution**.
>
> - In addition to the proposed neural model NC-GNN, **our proposed NC-1-WL for graph isomorphism test is also a contribution**. NC-1-WL can replace 1-WL as a more powerful graph isomorphism test algorithm. In comparison, previous expressive GNNs, including subgraph GNNs, only consider neural version for graph learning, while we also deliver a better algorithm for the graph isomorphism problem.
>
> > It is just a special case of recently NeurIPS 2022 paper [How Powerful are K-hop Message Passing Graph Neural Networks], with K=1. The NeurIPS paper is general and with extensive theoretical analysis. In this paper I can only find the simplicity as a contribution. The incremental design doesn't help the research area.
> - Note that the NeurIPS2022 accepted papers are released on 9/19/2022, which is just one week before the ICLR2023 submission deadline.
> - After a careful reading, we agree that our work is related to this NeurIPS2022 paper. However, there are still key differences. (1) As mentioned before, we first propose the NC-1-WL as a deterministic algorithm for the graph isomorphism problem, which has not been considered by the NeurIPS2022 paper. (2) The NeurIPS2022 paper mainly focuses on formulating the K-hop message passing framework and analyzing its expressive power. In contrast, we dedicate to the consideration of edges among neighbors, leading to the powerful and efficient NC-1-WL and NC-GNN. (3) The NeurIPS2022 paper considers the subgraph information by encoding the number of peripheral edges, while we are modeling the communication among neighbors (in other words, feature interaction among neighbors) (i.e., the 3rd term in Eq. (6)). With our dedicatedly designed model, our proof of expressivity is totally different from previous work. These three differences make our work to be novel. We have clarified these differences in our revision.
>
>
>
> > All theorems are mentioned in existing papers. The 3-WL bound is inside [Understanding and extending subgraph gnns by rethinking their symmetries].
> - The 3-WL bound proof in the mentioned paper is for subgraph GNNs. Since our specific design is different from subgraph GNNs (as described in our first response), our proof for expressiveness applies totally different techniques compared to previous work. Our detailed proof in Appendix A.2 can verify this claim.
>
> - We respectfully disagree that “proving the same upper bound for different methods” equals to “all theorems are mentioned in existing papers”. Many previous expressive GNN works have mentioned and proved the 3-WL upper bound for their corresponding approaches. The key point is that **our approach is different and our corresponding proof is new**.
>
> ---> The remaining response is in the following part 2 --->

---

### Comment · Reviewer_Agxt · 2022-11-24
**Regarding the way of picking datasets**

From the author's response, I have noticed that the author used #Message_NC as a metric to select datasets to evaluate, and I would like to emphasize that this is not the correct way to evaluate your method, and it also shows significant shortcomming of the proposed algorithm.

1. The author claim that if #message_NC is 0, then the proposed method will be equivalent to GIN, however the proposed method is not as efficient as GIN under this case.
2. Based on the statistics reported, there are a large amount of datasets that having #message_NC < 0.2, and the author said the result is similar to GIN on these dataset (without detailed number reported). However many existing subgraph-based GNN still shows significant performance improvement over these datasets, like ZINC with #message_NC only being 0.0008. These subgraph-based GNN shows the great performance improvement with MAE improves from 0.016 to 0.008.
3. Another theoretical problem, is that the proposed method **cannot count cycles**. As we know that cycles are important substructure feature for molecular dataset, the proposed method basically won't work for molecular datasets. In fact, for any substructure with 0 #message_NC, the proposed method cannot count it.

All these show a significant shortcoming of the proposed method, and the way of picking evaluation dataset clearly avoids talking all these shortcomings.

---

> ### Author Response · Authors · 2022-11-26
> **Clarified the misunderstanding of the complexity and previous experiments; recap our claims.**
>
> As pointed out in [our previous response](https://openreview.net/forum?id=DT7btGps59z&noteId=62aQ5aKsao), **we do have clearly clarified the use case of our method** and **empirically demonstrated the effectiveness of our method in the desirable use cases**. Specifically, if there are edges among neighbors, our NC-GNN serves as a more **powerful** method than original massage passing GNNs (such as GIN), while being more **efficient** than most expressive GNNs. These main claims have been **supported by our existing experimental directly**. As this has been made clear, we provide response to the additional comments below.
>
> > The author claim that if #message_NC is 0, then the proposed method will be equivalent to GIN, however the proposed method is not as efficient as GIN under this case.
>
> `Our NC-GNN does have the same efficiency as GIN if #message_NC is 0`. As described in our paper (following Eq. (6)), if #message_NC is 0, then the proposed method will **reduce** to GIN. In other words, our NC-GNN will become GIN **exactly**. We also analyzed the complexity of our NC-GNN in Sec 4 and Table 1, both the memory and time complexity will become the same as GIN if #message_NC is 0. Note that for the memory complexity of NC-GNN, we can choose to additionally store the representations for edges among neighbors. If #message_NC is 0, this additional memory usage will also be 0.
>
> > Based on the statistics reported, there are a large amount of datasets that having #message_NC < 0.2, and the author said the result is similar to GIN on these dataset (without detailed number reported). However many existing subgraph-based GNN still shows significant performance improvement over these datasets, like ZINC with #message_NC only being 0.0008. These subgraph-based GNN shows the great performance improvement with MAE improves from 0.016 to 0.008.
>
> Since we dedicated to considering the edges among neighbors (i.e., one-hop neighbors), it is reasonable to compare our method to subgraph-based GNNs with one-hop subgraph. As shown in Table 6 in the [GNN-AK paper](https://openreview.net/forum?id=Mspk_WYKoEH), the one-hop subgraph GNN-AK only improves the performance on ZINC from 0.163 (GIN) to 0.147. Note that GNN-AK also uses additional Distance-to-Centroid encoding compared to GIN. As shown in its Table 7, the improvement brought by Distance-to-Centroid encoding is obvious. This means that the improvement brought by considering one-hop subgraph information on ZINC is not obvious. **These empirical observations from GNN-AK also support our claim that if #message_NC is low, the performance improvement by considering one-hop subgraph information is incremental (or not obvious)**. This intuitively makes sense, since GIN can also consider almost all the information in the one-hop subgraphs if there are few edges among neighbors.
>
> > Another theoretical problem, is that the proposed method cannot count cycles. As we know that cycles are important substructure feature for molecular dataset, the proposed method basically won't work for molecular datasets. In fact, for any substructure with 0 #message_NC, the proposed method cannot count it.
>
> * Again, our claim is that our NC-GNN is a powerful and efficient model by considering the edges among neighbors in the one-hop subgraph. It reaches a sweet spot between expressivity and scalability, which also have been demonstrated by our experiments.
>
> * **Counting cycles is a specific task and out of the scope of our work**. For molecules without triangles, our NC-GNN reduces to GIN. Overall, we expect our NC-GNN to be a strong and efficient basic model for social network graphs, interaction-based graphs (such as protein-protein interaction), etc.

---

### Decision · Program_Chairs · 2023-01-20

**Decision:**

Reject

**Justification For Why Not Higher Score:**

N/A

**Justification For Why Not Lower Score:**

N/A

**Metareview: Summary, Strengths And Weaknesses:**

Thank you for submitting your work to ICLR 2023. This work stirred a lot of discussions. Eventually it did not receive sufficient support from any of the reviewers to grant accepting it this time.
There are several issues raised by reviewers and discussed by authors: (i) the incremental contribution of adding the multiset of 1-ring edges; (ii) completeness and quality of experiments; and (iii) whether a very related arxiv paper published May 26 is to be considered prior work, and the authors' claims for misconduct.

Regarding (i) and (ii) the authors failed to convince the majority of the reviewers, that held to their original position. The idea of adding the multiset of 1-ring edges to the node's features is already very close to explored ideas in this space of strengthening the expressive power of MPNN. This alone, prevents us from recommending to accept this paper.

Regarding (iii), after discussion with the SAC and conference chairs, we think the authors' claims are unjustified. The guidelines explicitly say that authors *may* be excused for not knowing....and that reviewers are encouraged to used their own judgment. This doesn't give carte blanche to ignore arXiv paper. The said paper was posted on arXiv on May 26, 2022 more than four months before ICLR deadline, and we feel in this case the reviewer position is reasonable.